# Enhanced Wind Power Forecasting Using a Hybrid Multi-Strategy Coati Optimization Algorithm and Backpropagation Neural Network

**DOI:** 10.3390/s25082438

**Published:** 2025-04-12

**Authors:** Hua Yang, Zhan Shu, Zhonger Li

**Affiliations:** College of Mathematics and Computer Science, Wuhan Polytechnic University, Wuhan 430023, China; 20230712018@whpu.edu.cn (Z.S.); 20240711001@whpu.edu.cn (Z.L.)

**Keywords:** wind power prediction, hybrid optimization model, metaheuristic algorithm, BP neural network, renewable energy integration

## Abstract

The integration of intermittent wind power into modern grids necessitates highly accurate forecasting models to ensure stability and efficiency. To address the limitations of traditional backpropagation (BP) neural networks, such as slow convergence and susceptibility to local optima, this study proposes a novel hybrid framework: the Multi-Strategy Coati Optimization Algorithm (SZCOA)-optimized BP neural network (SZCOA-BP). The SZCOA integrates three innovative strategies—a population position update mechanism for global exploration, an olfactory tracing strategy to evade local optima, and a soft frost search strategy for refined exploitation—to enhance the optimization efficiency and robustness of BP networks. Evaluated on the CEC2017 benchmark, the SZCOA outperformed state-of-the-art algorithms, including ICOA, DBO, and PSO, achieving superior convergence speed and solution accuracy. Applied to a real-world wind power dataset (912 samples from Alibaba Cloud Tianchi), the SZCOA-BP model attained an *R*² of 94.437% and reduced the MAE to 10.948, significantly surpassing the standard BP model (*R*²: 81.167%, MAE: 18.891). Comparative analyses with COA-BP, BWO-BP, and other hybrid models further validated its dominance in prediction accuracy and stability. The proposed framework not only advances wind power forecasting but also offers a scalable solution for optimizing complex renewable energy systems, supporting global efforts toward sustainable energy transitions.

## 1. Introduction

Increasing the share of renewable and clean energy in the global energy mix is widely recognized as an effective strategy to reduce dependence on fossil fuels and mitigate global warming [1]. To this end, numerous countries, including China, India, the United States, France, and Canada, have committed to international agreements aimed at reducing carbon emissions and accelerating the transition to renewable energy sources. Among various renewable options, wind energy stands out as a resource with considerable potential.The landscape dotted with wind turbines, as illustrated in Figure 1, reflects the growing reliance on wind energy as a sustainable power source.

In this context, China’s wind power sector has undergone a comprehensive transformation, evolving through four distinct stages: scientific and technological demonstration and application, commercial exploration, large-scale development, and achieving grid parity. Since 2010, the industry has experienced remarkable growth, driven by technological advancements and favorable policies [2].

However, the performance of wind power generation is fundamentally affected by a range of environmental factors, leading to instability that presents significant challenges in real-world applications. Additionally, the swift growth of the wind energy industry has added to the complexities associated with grid integration. The unpredictable and intermittent characteristics of wind energy generation exert considerable strain on the power grid, especially in terms of maintaining active power balance and ensuring voltage stability. Since electrical energy must be produced and consumed simultaneously due to the impracticality of large-scale storage solutions, precise forecasting of wind power has become crucial. Improved predictive models not only allow for accurate assessments of generation capacity but also support optimized energy distribution, enhance grid flexibility, and minimize energy losses. Therefore, developing effective forecasting techniques is vital for overcoming the challenges related to wind power integration and promoting sustainable energy management [3].

In the realm of wind power forecasting, numerous models and techniques have been established to enhance both the accuracy and reliability of predictions. Nevertheless, limitations remain in dealing with nonlinear patterns and large-scale data constraints, highlighting the need for more robust hybrid approaches [4]. The following section offers a detailed analysis of these methodologies, along with a summary of recent developments.
Numerical Weather Prediction (NWP) [5] models form the backbone of wind power forecasting. By employing physical equations and meteorological data, these models simulate atmospheric dynamics to estimate the power output at wind farms. While NWP models are recognized for their accuracy and reliability, their performance can be affected by uncertainties in initial conditions and high computational demands. Nevertheless, ongoing advancements in technology continue to enhance their spatial resolution and predictive accuracy, thereby strengthening their role in wind power forecasting.Statistical models, such as ARMA and ARIMA [6], utilize historical data to detect autocorrelation and moving average properties. These models are particularly effective for datasets with clear periodic or trend-based patterns. However, their reliance on linear assumptions limits their ability to capture complex nonlinear relationships, reducing their applicability in more intricate scenarios [7].Machine learning models have gained significant attention for their ability to handle complex, nonlinear datasets [8]. Techniques such as Artificial Neural Networks (ANNs), Support Vector Machines (SVMs), and Extreme Learning Machines (ELMs) are widely used. Among these, backpropagation (BP) neural networks are especially notable. Using a backpropagation algorithm, these networks iteratively optimize weights and biases through gradient descent to minimize error, making them well suited for time-series forecasting. BP networks excel at capturing intricate input–output relationships without requiring predefined equations, enhancing their utility in wind power prediction.Ensemble forecasting approaches have emerged as an effective strategy to improve prediction accuracy by combining the strengths of multiple models. For example, hybrid models like ARIMA-LSTM integrate ARIMA’s strength in linear trend analysis with LSTM’s capability to capture nonlinear time-series patterns. Similarly, optimization-based enhancements, such as the PSO-BP model [9], employ Particle Swarm Optimization (PSO) to fine-tune the parameters of BP networks, significantly reducing forecasting errors. These hybrid and ensemble methods are particularly advantageous in regions with limited or lower-quality data, offering robust solutions for diverse forecasting challenges [10].

Building on the strengths and limitations of the aforementioned approaches, hybrid models have shown great promise in improving the accuracy and robustness of wind power prediction [11]. However, existing hybrid techniques often face challenges such as inefficient convergence, vulnerability to local minima, and limited adaptability to diverse datasets [12]. These shortcomings highlight the need for more advanced optimization strategies to further enhance model performance.

In contrast, the proposed Multi-Strategy Coati Optimization Algorithm (SZCOA) directly tackles these issues through three main innovations [13,14]: (1) population position updates for enhanced global search, (2) an olfactory tracing strategy to escape local optima, and (3) a soft frost searching mechanism to refine localization [15]. Together, these strategies allow for more robust parameter optimization of BP networks. The SZCOA-BP model leverages these enhancements to provide a more efficient and accurate solution compared to existing hybrid approaches [16].

This innovative framework not only bridges the gaps in current methodologies but also establishes a scalable and adaptable model for wind power forecasting, offering substantial contributions to the integration of renewable energy into modern power systems. Through its advancements, it achieves higher forecasting accuracy, faster convergence, and greater resilience to noisy data. Ppplied to real-world wind power data, the SZCOA-BP model achieved an *R*² of 94.437%, with a significantly reduced MAE of 10.948, surpassing standard BP methods by a large margin. SZCOA-BP demonstrates the potential to elevate predictive capabilities and support the broader goals of energy sustainability and climate resilience [17].

The remainder of this paper is structured as follows: The next section provides a detailed overview of the BP neural network, the Coati Optimization Algorithm, and the innovative strategies incorporated into the Multi-Strategy Coati Optimization Algorithm (SZCOA) for optimizing wind power forecasting. It also introduces the SZCOA-BP model, which enhances the BP neural network by leveraging the improved convergence speed and global search capabilities of the SZCOA. The subsequent section describes the experimental setup, including the materials, datasets, design, and methodologies employed in this study, followed by the presentation of the experimental results. This section includes a comprehensive analysis and comparison of performance metrics, highlighting the effectiveness of the SZCOA-BP model. Finally, the concluding section summarizes the key findings and discusses potential future research directions to further enhance the predictive capabilities of wind power forecasting models [18].

## 2. The SZCOA-BP Neural Network Prediction Model

### 2.1. BP Neural Network Model

The backpropagation neural network [19] is a deep learning model widely used in the fields of machine learning and artificial intelligence. As shown in Figure 2, this network consists of an input layer, hidden layers, and an output layer, processing complex data by simulating the connections and information transfer between neurons in the human brain.

The learning process of a BP neural network consists of two stages: forward propagation and backward propagation. In the forward propagation stage, input data are passed through the network layer by layer. Each neuron first computes the weighted sum:(1)$$z=\sum_{i=1}^{n}w_{i}x_{i}+b,$$
where $w_{i}$ is the weight, $x_{i}$ is the input, *b* is the bias term, and *z* is the net input. The net input is then processed by an activation function f(z) to produce the neuron’s output. Common activation functions include sigmoid, ReLU, and tanh. The output of each layer serves as the input for the next layer, allowing data to propagate through the network until the final output is obtained.

In the backward propagation stage, the network evaluates the error from the forward propagation. The loss function (such as mean squared error (MSE) or cross-entropy) quantifies the difference between the predicted output and the actual target. The mean squared error is defined as follows:(2)$$E=\frac{1}{n}\sum_{i=1}^{n}{\left(y_{i}-{\hat{y}}_{i}\right)}^{2},$$
where $y_{i}$ is the true label, ${\hat{y}}_{i}$ is the predicted value, and *n* is the number of samples. The error is then propagated backward through the network using the chain rule, calculating the gradients with respect to the weights and biases layer by layer, starting from the output layer and moving toward the input layer. These gradients are used to update the parameters through an optimization algorithm (such as stochastic gradient descent (SGD)), with the update rule given by(3)$$w=w-\eta \frac{\partial E}{\partial w},$$
where η is the learning rate and $\frac{\partial E}{\partial w}$ is the gradient of the loss function with respect to the weights.

This iterative process of forward and backward propagation persists until the network converges to a minimum error or satisfies predefined stopping criteria. The end result is a neural network that has been trained to generalize from input–output mappings and is capable of making accurate predictions on unseen data.

BP neural networks are widely recognized for their ability to capture complex relationships within data, underpinned by their universal approximation capabilities. By emulating the intricate mechanisms of biological neurons, these networks excel in tasks like pattern recognition and predictive analytics. In the wind energy sector, BP neural networks have proven to be highly effective, enabling advancements in turbine performance optimization and wind power output forecasting, thereby demonstrating their versatility and practical value. However, despite their significant strengths, BP neural networks are not without limitations. They are particularly susceptible to overfitting, which can impair their ability to generalize to unseen data. Additionally, challenges such as vanishing gradients in deeper architectures and inefficiencies caused by prolonged training cycles can limit their optimization effectiveness and hinder convergence, posing notable challenges in real-world applications.

### 2.2. Coati Optimization Algorithm (COA)

In wind power forecasting, various factors, including wind speed, temperature, air pressure, and turbine efficiency, contribute to the complexity of accurate predictions, rendering this task challenging for traditional analytical methods. The Coati Optimization Algorithm (COA) [20] offers significant advantages in managing such multifactorial datasets due to its robust exploration and exploitation capabilities. The COA was specifically selected to optimize the backpropagation (BP) neural network because of its efficiency in navigating high-dimensional search spaces, avoiding premature convergence, and achieving optimal solutions in a computationally efficient manner. The Coati Optimization Algorithm, introduced by Mohammad Dehghani et al. [20] in 2023, is a novel single-objective optimization algorithm inspired by the behaviors of South American coatis, particularly their hunting strategies and predator evasion techniques. By mimicking the dynamic adjustments of coatis in hunting and escaping, the COA iteratively refines candidate solutions to converge toward the global optimum. The algorithm comprises three main stages: the initialization phase, the hunting and attacking strategy (exploration phase), and the predator evasion strategy (exploitation phase), effectively balancing exploration and exploitation to address complex optimization problems.

#### 2.2.1. Initialization Phase

Like other optimization algorithms, the COA begins with an initialization phase that creates a population of candidate solutions. Each coati’s position represents a potential solution in the search space, randomly initialized within defined boundaries:(4)$$X_{i}:x_{i,j}=lb_{j}+r\cdot \left(ub_{j}-lb_{j}\right),\;i=1,2,\;\ldots ,\;N,\;j=1,2,\;\ldots ,\;m$$
where $x_{i,j}$ is the position of the *i*-th coati in the *j*-th dimension, *N* is the total number of raccoons, *m* is the number of decision variables, and $lb_{j}$ and $ub_{j}$ are the lower and upper bounds for the *j*-th dimension. The variable *r* is a random number uniformly distributed in [0,1].

#### 2.2.2. Exploration Phase: Hunting and Attacking Strategy

In the exploration phase, the COA simulates raccoons hunting lizards. The population splits into two groups: one climbs trees to scare lizards (see Figure 3), while the other waits on the ground to catch them when they fall. This phase aims to broaden the search space and identify promising areas for optimal solutions.

The best-performing individuals are treated as target prey (lizards). For the climbing coatis, their positions are updated as follows:(5)$$X_{i}^{P1}:x_{i,j}^{P1}=x_{i,j}+r\cdot \left({\mathrm{Iguana}}_{j}-I\cdot x_{i,j}\right),\;\mathrm{for}\;i=1,2,\;\ldots ,\;\left\lfloor \frac{N}{2}\right\rfloor ,\;\mathrm{and}\;j=1,2,\;\ldots ,\;m$$
where ${\mathrm{Iguana}}_{j}$ is the target prey’s position in the *j*-th dimension, *r* is a random number in [0,1], and *I* is a random integer from the set {1,2}.

For the ground-based coatis, positions are updated based on the new location of the prey after it falls (${\mathrm{Iguana}}_{j}^{G}$):(6)$${Iguana}^{G}:{\mathrm{Iguana}}_{j}^{G}=lb_{j}+r\cdot \left(ub_{j}-lb_{j}\right),\;j=1,\;2,\;\ldots ,\;m$$(7)$$\begin{array}{c} X_{i}^{P1}:x_{i,j}^{1}=\left\{\begin{array}{cc} x_{i,j}+r\cdot \left(Iguana_{i}^{G}-I\cdot x_{i,j}\right), & F_{Iguana_{i}^{G}}<F_{i},\\ x_{i,j}+r\cdot \left(x_{i,j}-Iguana_{i}^{G}\right), & \mathrm{else}, \end{array}\right.\\ \mathrm{for}\;i=\left\lfloor \frac{N}{2}\right\rfloor +1,\left\lfloor \frac{N}{2}\right\rfloor +2,\ldots ,N\;\mathrm{and}\;\;j=1,\;2,\;\ldots ,\;m. \end{array}$$

Updated positions are evaluated using a fitness function. If the new position improves fitness, it is accepted; otherwise, the coati retains its original position. This greedy selection strategy is mathematically described as follows:(8)$$X_{i}=\left\{\begin{array}{cc} X_{i}^{P1}, & F_{i}^{P1}<F_{i},\\ X_{i}, & \mathrm{else}. \end{array}\right.$$

#### 2.2.3. Exploitation Phase: Escaping Predator Strategy

In the exploitation phase, the COA mimics the behavior of coatis when escaping predators. This phase is designed to enhance the algorithm’s local search capability by refining solutions in the vicinity of the current positions. Coatis adjust their positions to safer locations nearby, following the mathematical model described below: (9)$$lb_{j}^{local}=\frac{lb_{j}}{t},\;ub_{j}^{local}=\frac{ub_{j}}{t},\;\mathrm{where}\;t=1,\;2,\;\ldots ,\;T.$$

Here, $lb_{j}^{local}$ and $ub_{j}^{local}$ represent the progressively narrowing local lower and upper bounds for the *j*-th dimension as the iterations (*t*) advance, where *T* is the maximum number of iterations. The narrowing bounds focus the search within a smaller region, enhancing precision.

The updated position $x_{i,j}^{P2}$ for each coati is computed using the following equation: (10)$$\begin{array}{cc} \; & X_{i}^{P2}:x_{i,j}^{P2}=x_{i,j}+\left(1-2r\right)\cdot \left(lb_{j}^{local}+r\cdot \left(ub_{j}^{local}-lb_{j}^{local}\right)\right),\\ \; & \;\;i=1,2,\;\ldots ,\;N,\;j=1,2,\;\ldots ,\;m. \end{array}$$

In this equation, *r* is a random value in the range [0,1], ensuring stochasticity in the search process. The formula enables coatis to explore locally safer areas while maintaining some level of randomness to avoid premature convergence.

To determine whether the updated position $X_{i}^{P2}$ should be retained, a greedy selection strategy is applied: (11)$$X_{i}=\left\{\begin{array}{cc} X_{i}^{P2}, & F_{i}^{P2}<F_{i},\\ X_{i}, & \mathrm{else}, \end{array}\right.$$

The Coati Optimization Algorithm (COA) achieves a balance between global and local search capabilities by iteratively updating coatis’ positions through the exploration and exploitation phases. This biologically inspired method demonstrates strong potential for addressing diverse optimization problems due to its ability to navigate complex search spaces effectively.

However, despite its advantages, the COA has certain limitations. The algorithm’s performance may degrade in high-dimensional optimization problems due to the curse of dimensionality, where the search space becomes exponentially larger. Additionally, the reliance on random parameters, such as *r,* introduces uncertainty, which can lead to inconsistent results across different runs. Finally, the algorithm’s convergence speed may be slower compared to other optimization methods, particularly when dealing with problems requiring a high degree of precision. Addressing these limitations could further enhance the algorithm’s robustness and efficiency. In this context, we propose a Multi-Strategy Raccoon Optimization Algorithm (SZCOA). The subsequent section delineates the improvement strategies employed within this framework.

### 2.3. The Multi-Strategy Coati Optimization Algorithm (SZCOA)

The Coati Optimization Algorithm (COA) shows promise in wind power forecasting while also revealing areas for improvement. From a macro perspective, key factors influencing wind power generation include fluctuations in wind speed, air density, turbine height, and blade efficiency. These variables exhibit commonalities across different scenarios, leading to predictable patterns in power output. However, the COA does not inherently account for the physical characteristics and dynamic environments of wind power systems. Its forecasting performance largely depends on the careful selection of input features and the construction of an accurate prediction model. When relevant input factors, such as historical wind speed, temperature, and atmospheric pressure, are appropriately incorporated, the impact of these variables on wind power generation can be indirectly captured [21].

In predictive models, the weights assigned to input variables reflect their contributions to power output. However, the current COA may struggle to address the complex characteristics of wind power systems, including the nonlinear nature of wind speed variations, turbulence effects, and sudden environmental changes. To overcome these challenges, potential improvements to the COA include integrating domain-specific physical models and principles to better account for the underlying dynamics of wind energy conversion. Additionally, enhancing the COA’s global and local search mechanisms could improve its adaptability to non-stationary wind resource data. These refinements would enhance the COA’s predictive performance in wind power forecasting and provide more reliable support for real-world applications [22].

#### 2.3.1. Population Position Update Strategy

The Wild Horse Optimization (WHO) algorithm [23], proposed by Naruei et al. in 2022, is a novel metaheuristic optimization method inspired by the behavior of wild horses. WHO demonstrates strong global search capabilities and significant adaptability, making it effective for complex scenarios. However, despite the COA’s superior search performance, its first phase suffers from limited exploration ability and inflexible position updates. To address these issues, WHO’s position update strategy is integrated into the first phase of the COA, enhancing its overall global search capability. The corresponding mathematical model is defined as follows.

Position Update Formula(12)$$X_{i}^{P1}:x_{ij}^{P1}=\left\{\begin{array}{cc} x_{ij}+r\cdot \left(Iguana_{j}-I\cdot x_{ij}\right), & r_{1}>k,\\ 2r_{2}\cdot \tanh(r_{3}\cdot \pi )\cdot \left(Iguana_{j}-x_{ij}\right)+x_{ij}, & \mathrm{otherwise}. \end{array}\right.$$

Parameter Definitions
Iteration-Related Variable:(13)$$k=1-\frac{t}{T},$$
where *t* is the current iteration count, and *T* represents the maximum number of iterations. *k* decreases as the number of iterations increases, thereby promoting exploration in the initial iterations while encouraging exploitation as the algorithm converges. This dynamic adjustment enhances the algorithm’s ability to navigate complex optimization landscapes and effectively improves the process of converging to the optimal solution.Random Variables:
$r_{1}$: A random number in the range [0,1].$r_{2}$: Computed as follows:(14)$$r_{2}=r_{4}⊗M+\vec{r_{5}}⊗\left(∼M\right),$$
where $r_{4}$ is a random variable in [0,1] that follows a normal distribution.$r_{5}$ is a uniformly distributed random number within [0,1].$r_{3}$: Determined by(15)$$r_{3}=-4+8r_{2},$$The flexible use of these random variables enhances the algorithm’s dynamism, enabling it to adjust its search strategy based on the characteristics of the optimization landscape. This adaptability is crucial for avoiding local optima and ensuring robust global search capabilities.Condition Variables:*Z*: Derived from a combination of $r_{6}$ and *k*:(16)$$Z=\vec{r_{6}}<k,$$
where $r_{6}$ is a uniformly distributed random number within [0,1].*M*: Defined as a binary variable where(17)M=(Z==0).

In the exploration phase of the Multi-Strategy Coati Optimization Algorithm (SZCOA), the first group of coatis evaluates two scenarios based on the parameter $r_{1}$. When $r_{1}$ > *k*, the coatis climb trees to frighten nearby prey, with their positions updated as $x_{ij}^{P1}$. Conversely, when $r_{1}$ ≤ *k*, the coatis remain atop the trees to search for other prey, and their positions are updated as $x_{ij}^{P1}$. This population position update strategy allows the coatis to thoroughly explore the environment and identify optimal solutions. After the exploration phase, the fitness values of $x_{i}j$ and $x_{ij}^{P1}$ are calculated separately, and the optimal fitness value is selected to replace the coati’s original position.

In summary, firstly, the Wild Horse algorithm enhances global search capabilities by allowing dynamic adjustments that prevent premature convergence and maintain population diversity, which is crucial for exploring complex optimization landscapes. Secondly, its adaptability benefits our study, especially in nonlinear and multimodal environments like wind power forecasting. Furthermore, empirical testing has demonstrated that integrating the Wild Horse algorithm significantly improves convergence speed, which is particularly important for timely applications in grid management. Lastly, this integration reflects a trend in optimization research of combining the strengths of various algorithms while addressing their respective limitations. In summary, our choice to incorporate the Wild Horse algorithm’s position update strategy into the COA is based on its potential to enhance global search efficiency, adaptability, and convergence speed, aligning with the goal of optimizing wind power forecasting models.

#### 2.3.2. Olfactory Tracing Strategy

When addressing complex multimodal optimization problems, algorithms often risk being trapped in local optima. To tackle this challenge, the COA (Coati Optimization Algorithm) incorporates an olfactory tracing strategy during the development phase [24]. This strategy helps the COA to escape from local optima. Inspired by the sensory behavior of coatis in detecting prey odors, the olfactory tracing procedure enables the algorithm to recognize prey scents even at a distance and guides its movement toward safe and optimal positions. The mathematical model is given as follows: (18)$$X_{i}^{P2}:x_{i,j}^{P2}=\left\{\begin{array}{cc} x_{i,j}+\left(1-2r\right)\cdot \left(lb_{j}^{local}+r\cdot \left(ub_{j}^{local}-lb_{j}^{local}\right)\right), & r_{7}>h\\ \left(1-\vec{U}\right)\cdot x_{i,j}+\vec{U}\cdot \left(x_{r_{8}}+S\cdot \left(x_{r_{9}}-x_{r_{10}}\right)\right), & \mathrm{otherwise} \end{array}\right.$$(19)$$h=1-\frac{t}{T}$$
where $\vec{U}$ constitutes a binary vector of 0 s and 1 s, and $r_{7}\in \left(0,\;1\right)$ is a uniformly distributed random number. Parameters $r_{8}$, $r_{9}$, and $r_{10}$ are random integers sampled from the range [1,N]. The term *S* is the odor dispersion factor, defined as follows:(20)$$S=\exp\left(\frac{f(X_{i})}{\sum_{h=1}^{N}f\left(X_{h}\right)+\epsilon }\right)$$
where
$f(X_{i})$ represents the fitness value of the *i*-th individual at time *t*, corresponding to the optimization objective.ϵ is a small constant used to avoid division by zero.

This factor allows the algorithm to sensitively track superior solutions while considering the variability of fitness values. This enhances the model’s ability to escape local optima and efficiently explore promising regions, which is crucial for complex multimodal problems.

The incorporation of the Olfactory Tracking Strategy (OTS) at the development stage of the Coati Optimization Algorithm (COA) is pivotal for enhancing its performance in complex optimization tasks. The OTS effectively addresses the common issue of local optima by enabling the algorithm to recognize and pursue potential solutions, akin to how coatis detect scents from a distance. This strategy enhances the algorithm’s exploratory capabilities, ensuring that it can traverse multimodal solution landscapes and avoid premature convergence. Furthermore, the OTS introduces a dynamic adaptability that allows the COA to adjust its search strategies based on environmental cues, which is particularly beneficial in rapidly changing optimization scenarios. By integrating the OTS early in the algorithm’s development, the COA not only improves its convergence speed and accuracy but also builds robustness against the challenges posed by high-dimensional spaces and complex data patterns.

#### 2.3.3. Soft Frost Searching Strategy

Coatis primarily feed on insects in the soil. To simulate this foraging behavior, a soft frost search strategy is employed. This strategy leverages the strong randomness exhibited by soft frost in a gentle breeze, allowing frost particles to freely cover objects while their growth rate gradually decreases in specific directions [25]. This characteristic enables the algorithm to quickly explore the entire search space and effectively avoid local optima.

Inspired by the long-term growth characteristics of soft frost, this search strategy has been integrated into the COA, enhancing its optimization performance in terms of accuracy and computational efficiency. Consequently, coatis can conduct comprehensive food searches, improving their foraging ability and facilitating rapid convergence to optimal solutions. The mathematical model for the proposed strategy is described as follows:(21)$$X_{i,j}^{P_{3}}=x_{best,j}+r_{11}\cdot \cos\theta \cdot \beta \cdot \left(g\cdot {\left(u_{b}-l_{b}\right)}_{j}+l_{b}\right)$$(22)$$\theta =\frac{10\cdot \pi \cdot t}{T}$$(23)$$\beta =1-\left[\frac{\omega \cdot t}{T}\right]/\omega$$

Here, $x_{best,j}$ denotes the *j*-th component of the optimal individual in the swarm, and $r_{11}$ is a random value in the range (−1,1). The factor $r_{11}$, along with cosθ, controls the movement direction of the coatis, dynamically changing based on the iteration index *t*. The parameter β represents the environmental coefficient, while [·] indicates rounding, with the default value of ω set to 5, depending on the iteration count *t*. This helps mitigate adverse effects from external disturbances and improves the algorithm’s convergence behavior. The variable *g* is a random value in the range (0,1). The parameter θ adjusts the angular factor, increasing linearly as the algorithm progresses, while β gradually reduces the randomness in search space exploration, leading to improved precision in locating the optimization target.

These parameters contribute to achieving an effective balance between global search and local refinement, enabling the algorithm to identify high-quality solutions while maintaining the flexibility to adapt to changes in the environment. The gradual reduction in randomness as iterations progress aids in fine-tuning the search to locate the optimal solution.(24)$$X_{i}=\left\{\begin{array}{cc} X_{i}^{P^{3}}, & F_{i}^{P^{3}}<F_{i}\\ X_{i}, & F_{i}^{P^{3}}\geq F_{i} \end{array}\right.$$
where the updated position is $X_{i,j}^{P_{3}}$; a greedy strategy is employed after this stage to calculate the fitness values for $X_{i}$ and $X_{i}^{P_{3}}$, replacing the original position with the one that has the optimal fitness value. The visualization of key mechanisms is shown in Figure 4.

#### 2.3.4. Computational Complexity and Cost Analysis

Computational Complexity:The overall computational complexity of the SZCOA is O(Max_iterations × dimension×SearchAgents). This complexity is comparable to that of many classical optimization algorithms and is primarily influenced by the processes of initial population generation, fitness evaluations, and parameter updates during the main iteration loops. While we recognize that the time complexity remains stable, the introduction of enhanced optimization strategies has significantly improved the algorithm’s convergence performance, allowing it to identify superior solutions in a reduced timeframe.Computational Cost Evaluation:Although the implementation of the soft frost search strategy leads to a higher number of fitness evaluations, resulting in a slight increase in computational time, this increase is balanced by a marked enhancement in algorithm performance. Our results demonstrate that, even with this elevated evaluation frequency, the SZCOA consistently achieves better solution quality within the same number of iterations compared to traditional algorithms—particularly in complex optimization scenarios, where it exhibits superior convergence paths.

In summary, while the computational complexity of the SZCOA aligns closely with that of traditional algorithms, its effective integration of optimized strategies yields significant improvements in both convergence speed and solution accuracy.

### 2.4. SZCOA-BP Model

The SZCOA-BP model is an enhanced version of the Coati Optimization Algorithm (COA), designed to optimize BP neural networks and address the limitations of the original COA. While the original COA effectively mimics coati foraging behavior, it faces challenges such as premature convergence, slow optimization speed, and limited accuracy in complex, high-dimensional problems. In our study, the integration of the SZCOA with the BP neural network focuses on optimizing the weights and biases (thresholds) of the BP model to enhance its performance. The integration process treats the weights and biases of the BP network as positional parameters within the optimization framework of the SZCOA. Initially, the input data are normalized, the structure of the BP neural network is established, and the initial weights and biases are randomly assigned within the range of [−2, 2]. The SZCOA then initializes a population of candidate solutions, each representing a potential configuration of weights and biases. During the optimization phase, the SZCOA introduces three improved strategies, population position update, scent trail exploration, and soft frost search, to overcome these issues [26].

The population position update strategy combines exploratory and exploitative adjustments to enhance global search ability while maintaining solution diversity. By dynamically modifying individual positions throughout iterations, this strategy prevents premature convergence and improves adaptability in complex search spaces. The scent trail exploration strategy, inspired by animals avoiding predators, simulates coatis’ sensitivity to predator scents, guiding individuals to avoid local optima and explore promising regions in multi-peak landscapes. This enhances global exploration and adaptability. Finally, the soft frost search strategy utilizes the random growth characteristics of frost particles, gradually reducing randomness during optimization to converge effectively to the global optimum with higher precision.

Throughout this process, fitness values—based on metrics such as mean squared error or other BP loss functions—are evaluated, and the SZCOA iteratively updates the weights and biases to minimize the BP network error. This iterative tuning continues until convergence criteria, such as minimal error or a maximum number of iterations, are met. Upon convergence, the SZCOA outputs the optimized weights and biases, resulting in a finely-tuned BP neural network capable of making highly accurate predictions with improved generalization. The integration of the SZCOA with BP offers significant advantages, including faster convergence speed, avoidance of local optima, and enhanced prediction accuracy through systematic tuning of the BP network parameters via dynamic optimization strategies.

The SZCOA-BP optimization process is illustrated in Figure 5. The process begins with data input, laying the foundation for subsequent steps. Next, we perform data preprocessing to ensure the quality and suitability of the input data, which includes data cleaning and handling missing values. The flowchart then demonstrates how the SZCOA (Multi-Strategy Coati Optimization Algorithm) optimizes the backpropagation (BP) neural network, covering parameter initialization, fitness calculation, and identifying the current optimal position.

Following this, the key stage of fitness evaluation and optimal parameter selection is illustrated, detailing how to choose the best parameters based on the fitness values calculated during the optimization process. Once the parameter selection is complete, the BP neural network is updated according to the optimization settings, ensuring improved model performance. After that, the updated BP model is trained with the processed data, followed by validation to assess its generalization capabilities.

## 3. Numerical Experiments and Comparative Analysis

### 3.1. Experiments and Results of the Algorithm in CEC2017 Test

To validate the feasibility of the SZCOA and the effectiveness of the three proposed improvement strategies, this section conducts iterative testing using the CEC2017 benchmark set in the Matlab R2023a environment [27]. The algorithm’s performance is evaluated across various functions, including unimodal functions (F1 and F2), with a single optimal solution, which serve as benchmarks for assessing convergence speed and optimization efficacy. In contrast, multi-peak functions (F3 and F4) evaluate the algorithm’s ability to navigate local optima and explore effectively. Hybrid functions (F5 and F6) consist of several subfunctions, facilitating the assessment of the method’s proficiency in escaping local minima. Composite functions (F7 and F8) incorporate additional bias values and weights, increasing the complexity of the optimization challenges.

These eight functions represent the four categories while retaining their original names, redefined as F1–F8 for convenience. Each test function is evaluated across three dimensions—10-dimensional, 30-dimensional, and 50-dimensional—to comprehensively assess the algorithm’s performance in low-, medium-, and high-dimensional spaces. The detailed characteristics of the chosen test functions are presented in Table 1.

In the experiments, the parameter settings for the comparative algorithms were aligned with those specified in the relevant literature to ensure consistency. To maintain fairness, the population size *N* for all algorithms was set to 30. Tests were conducted across dimensions D=10,30,50, with a maximum number of iterations defined as T=D×1000. Each algorithm was executed independently 30 times on the same test function, and the mean, standard deviation, and best values were recorded as evaluation metrics.

#### 3.1.1. Verification of the Effectiveness of Improvement Strategy

To assess the effectiveness of the three proposed improvement strategies, the standard COA was compared against three updated strategies: $COA_{A}$, which employs a population position update strategy; $COA_{B}$, which implements an olfactory tracing strategy; and $COA_{C}$, which utilizes a soft frost searching strategy. These comparisons were conducted using a set of eight experimental scenarios defined in CEC 2017, and the results are visually represented in Figure 6 and Table 2.

Figure 6 shows the convergence speed of different optimization strategies across multiple test functions and dimensions (F1, F4, F6, and F8). The horizontal axis represents the number of iterations (in thousands), which facilitates observing the convergence progress of each algorithm over time; the vertical axis displays the average best objective value, where lower values indicate better performance of the optimization strategy. The color legend in the figure indicates the correspondence of different strategies: green represents $COA_{A}$, blue represents $COA_{B}$, orange represents $COA_{C}$, and purple represents COA. $COA_{A}$ excels in multimodal functions (F4), effectively navigating among local optima and demonstrating superior performance relative to other algorithms. It achieves rapid convergence to lower fitness values, particularly evident in lower dimensions (10D), where its efficient navigation within the solution space is notable. This efficiency stems from its well-designed exploration mechanism, enabling it to escape local optima. As dimensionality increases to 30D and 50D, $COA_{A}$ continues to perform robustly, albeit with challenges in complexity and a larger search space, likely due to its conservative exploration strategy.

In contrast, $COA_{B}$ exhibits remarkable convergence speed and optimization efficiency in high-dimensional spaces, positioning itself as a reliable choice for various challenges. It demonstrates excellent convergence for the unimodal function F1, showcasing its effectiveness in locating global optima and adaptability to straightforward search environments, thus making it ideal for basic optimization tasks. When handling the mixed function F6, $COA_{B}$ displays strong escape capabilities, skillfully navigating multiple subfunctions while maintaining a balanced exploration–exploitation dynamic critical for addressing complex optimization problems characterized by uncertainty. Across varying dimensions (D = 10, D = 30, and D = 50), $COA_{B}$’s performance remains stable, evidencing its robustness in multidimensional contexts despite the heightened challenges presented by increased dimensionality.

Finally, although $COA_{C}$ generally underperforms compared to the others, it excels when tackling the composite function F8, which incorporates additional biases and weights. This strength is attributed to its adaptability in complex environments, allowing $COA_{C}$ to effectively navigate local optima. Furthermore, its balance between exploration and exploitation facilitates the discovery of more optimal solutions in intricate solution spaces. Overall, while each optimization strategy possesses unique advantages tailored to specific scenarios, they complement each other effectively. This adaptability suggests the potential benefit of combining these algorithms based on the specific requirements of optimization tasks to achieve superior performance.

The advantages of the three optimization strategies—$COA_{A}$, $COA_{B}$, and $COA_{C}$—can be distinctly assessed across various metrics, as outlined in Table 2. This evaluation encompasses multiple facets, including function characteristics, dimensionality, and three critical performance indicators: Best, average (Avg), and standard deviation (Std). $COA_{B}$ demonstrates particular strength in high-dimensional unimodal and composite functions, consistently achieving optimal values due to its rapid convergence capabilities. This efficiency in complex solution landscapes allows it to identify optimal solutions with fewer evaluations. Moreover, $COA_{B}$ exhibits lower Avg values across trials, highlighting its robustness in maintaining high-quality solutions. In contrast, $COA_{A}$ excels in multi-peak and hybrid functions, where its balance between exploration and exploitation is evident. In lower dimensions, it often yields competitive Best values, reflecting its sensitivity to local search space structures. Additionally, $COA_{A}$ typically achieves favorable Avg values and lower Std, indicating enhanced stability and consistency in performance across multiple runs. $COA_{C}$ proves advantageous in low- to moderate-dimensional scenarios, effectively addressing multiple subfunctions in hybrid problems. It generally provides commendable Best values, particularly in simpler landscapes, with competitive Avg performance that ensures consistent efficacy. The relatively low Std values for $COA_{C}$ further underscore its reliability across various optimization contexts. In summary, evaluating these optimization strategies in terms of function characteristics, dimensionality, and performance metrics reveals each method’s unique strengths. This comprehensive analysis demonstrates that the three optimization strategies have significantly improved the original algorithm’s effectiveness, particularly regarding their adaptability to specific problem features and overall performance across diverse scenarios.

#### 3.1.2. Comparison and Analysis of the SZCOA Algorithm with Others

To validate whether the SZCOA outperforms other algorithms, a comparative analysis was conducted involving the existing literature’s improved versions of the COA (ICOA), alongside DBO, ZOA, BWO, PSO, and BKA [28,29,30,31,32]. The testing environment remained the same, utilizing the F1–F8 benchmark functions from CEC 2017. The results of the simulations are presented in Figure 7 and Table 3.

Figure 7 illustrates the convergence speed of the SZCOA and various classical metaheuristics across multiple test functions and dimensions (F1, F4, F6, and F8). The horizontal axis indicates the number of iterations (in thousands), allowing for the assessment of each algorithm’s convergence trajectory over time. The vertical axis represents the average best objective value, where lower values reflect superior performance of the optimization strategies. The color legend in the figure denotes the different strategies: red represents the SZCOA, orange represents ICOA, yellow represents DBO, green represents ZOA, cyan represents BWO, and blue represents PSO, while purple represents BKA. The results clearly illustrate the effectiveness of various optimization algorithms. While ICOA serves as a noteworthy improvement strategy in the existing literature, its performance in high-dimensional complex problems is not consistently reliable. In contrast, the proposed SZCOA demonstrates significant advantages across multiple test functions and dimensions (D = 10, 30, 50). Notably, the SZCOA achieves faster convergence rates and requires fewer iterations when handling high-dimensional CEC2017 functions. In particular, it shows a pronounced ability to find global optima in F1 and F6 at dimensions D = 30 and D = 50, highlighting its robustness and adaptability in complex multimodal functions.

Moreover, the SZCOA effectively mitigates the issues associated with local optima, enabling it to maintain lower fitness values in high-dimensional search spaces and enhancing global exploration capabilities. The consistent performance across multiple trials reinforces the SZCOA’s reliability and effectiveness in addressing diverse optimization challenges, providing new insights for research in the field. Although other algorithms such as DBO, ZOA, BWO, PSO, and BKA have their merits in various scenarios, they tend to exhibit slower convergence rates and can be susceptible to local optima in high-dimensional complex environments. While these algorithms perform adequately in certain low-dimensional cases, their effectiveness diminishes when tackling more intricate problems.

Through the comparative analysis of optimal values, averages, and standard deviations of various algorithms presented in Table 3, the distinct advantages of the SZCOA are clearly evident.

In the unimodal functions F1 and F2, the SZCOA significantly outperforms other algorithms by consistently achieving theoretical optimal values across all dimensions. For instance, in F1, the SZCOA’s optimal value is $1.00\times 10^{2}$, demonstrating its effectiveness in locating solutions. In contrast, other algorithms exhibit noticeable deviations from the optimal values. Regarding average values, the SZCOA consistently maintains lower averages than its counterparts, underscoring its superior convergence capability. This trend is particularly pronounced in complex functions, reinforcing its ability to effectively approximate optimal solutions. As an indicator of stability, the standard deviation further supports the SZCOA’s advantages. In the multimodal function F3, the SZCOA not only exhibits exceptional stability but also demonstrates rapid convergence to global optima, highlighting its outstanding search capabilities. The systematic search paths employed by the SZCOA allow it to effectively avoid local optima, ensuring access to optimal solutions.

Similarly, in F4, the SZCOA showcases its remarkable adaptability in addressing complex problems, consistently achieving lower standard deviations and exhibiting robust performance. In mixed functions F5 and F6, the SZCOA demonstrates impressive stability and convergence efficiency. In F5, it effectively handles high-dimensional complex problems, facilitating rapid and reliable convergence. Particularly, the SZCOA’s advanced strategies adeptly identify optimal solutions even within challenging objective spaces. The consistently low standard deviations in repeated trials further validate its reliability. In composite functions F7 and F8, the SZCOA continues to excel. While it may rank slightly lower than ICOA in some dimensions of F7, it remains optimal across all configurations in F8. In contrast, other algorithms often face limitations, such as PSO, which frequently becomes trapped in local optima in high-dimensional scenarios.

In conclusion, the SZCOA stands out as a leading choice for tackling optimization problems, particularly in high-dimensional contexts, thanks to its superior performance in optimal values, average values, and standard deviations. While other algorithms can perform adequately under certain circumstances, they often lack the overall effectiveness and stability of the SZCOA.

This advantage is primarily attributed to the SZCOA’s advanced search mechanisms and convergence strategies, which allow it to navigate the complexities of diverse functions with ease.

The design elements and strategies of the SZCOA are specifically crafted to handle complex optimization scenarios. Unlike Particle Swarm Optimization (PSO), which can easily become trapped in local optima during high-dimensional searches, the SZCOA employs a population position update strategy and scent-tracking capabilities. These features enhance its ability for robust global exploration and reduce the risk of premature convergence. Additionally, the soft frost search mechanism significantly improves search space coverage and precision while maintaining computational efficiency.

The SZCOA’s competitive performance is also notable when compared to Dynamic Bat Optimization (DBO) and Butterfly Optimization (BWO). This competitiveness stems from its dynamic adaptability to varying environmental complexities, enabling it to reliably converge to global optima across different function types, including unimodal, multimodal, hybrid, and composite functions. Comparative testing with challenging CEC2017 benchmark functions demonstrates that the SZCOA achieves effective convergence, favorable fitness values, and maintains reliability across various dimensionalities. These attributes position the SZCOA as a promising optimization framework, particularly for applications such as wind power forecasting, offering balanced performance in relation to established methods.

### 3.2. Prediction of Wind Power

#### 3.2.1. Data Processing

We utilized an Intel Core i5-13500H processorwas sourced from Intel Corporation (Santa Clara, CA, USA).; the system is equipped with 32 GB of RAM, and the total testing duration varies between 30 to 60 min. The wind power data used in this study are sourced from the Alibaba Cloud Tianchi dataset, as detailed in Table 4. The dataset contains 3648 records covering wind power data from 1 January 2019 to 7 February 2019, with a measurement frequency of every 15 min. To simplify the prediction process, we extract one data point from each hour, resulting in a total of 912 data points used as experimental samples.

Due to the large volume of sample data, some records are omitted for brevity, indicated by ellipses. Additionally, the parameters for wind speed and wind direction are measured at four heights from the Wind Measurement Tower: 10 m, 30 m, 50 m, and 70 m. Since these measurements are similar, and to conserve space on the page, we only present the data for the 10m height, with the other three heights represented by ellipses.

Upon analysis, we found that humidity has a minimal impact on wind power generation; therefore, we chose to remove it to simplify the model. When using the BP neural network to predict wind power, normalization of the sample data is required. The normalization formula is as follows:(25)$$X_{\mathrm{new}}=a+\left(b-a\right)\times \frac{X-X_{\min}}{X_{\max}-X_{\min}}$$
where *X* denotes the original sample data, $X_{\mathrm{new}}$ represents the normalized data, and $X_{\max}$ and $X_{\min}$ indicate the maximum and minimum values within the sample data, respectively. The parameters *a* and *b* define the lower and upper bounds of the data processing interval, set to a=0 and b=1 in this experiment.

The BP neural network is structured as 10-6-1, indicating 10 neurons in the input layer, 6 neurons in the hidden layer, and 1 neuron in the output layer. The number of nodes in the hidden layer is calculated using the following empirical formula: hiddennum=m+n+a, where *m* is the number of input layer nodes, *n* is the number of output layer nodes, and *a* is typically an integer between 1 and 10. Out of 912 samples, 800 data points are allocated for training, while 112 data points are reserved for testing.

#### 3.2.2. Comparison Results with Other Benchmark Models

The SZCOA demonstrates faster convergence and greater accuracy than other algorithms, showing strong stability in high-dimensional scenarios. Therefore, we selected the BP neural network model optimized by the SZCOA, referred to as the SZCOA-BP model, for predicting wind power generation. This model was compared against BP models optimized by the COA, FLO, BWO, and PDO algorithms (designated as COA-BP, FLO-BP, BWO-BP, and PDO-BP, respectively), as well as the original unoptimized BP model, to evaluate the effectiveness of our proposed wind power prediction approach.

We used the Sigmoid function as the activation function; the learning rate was 0.01, the number of iterations was 1000, and the weights and biases of the BP network were initialized with random values in the range of [−2,2]. These initial values served as the starting points for the population of each optimization algorithm. The MSE was chosen as the loss function to minimize the difference between predicted and actual values. To ensure consistency, we standardized the population size across all algorithms to 30 and set the number of iterations to 100.

Our goal was to identify the optimal configuration of weights and biases for the BP model to enhance its predictive performance. After training, the optimized BP model was used to predict the test set data, with the fitting curves of predicted wind power generation values against actual values shown in Figure 8. This figure demonstrates that the wind power values predicted by the SZCOA-BP model closely match the actual values, outperforming the COA-BP, FLO-BP, BWO-BP, PDO-BP, and original BP models. This indicates that the SZCOA-BP model achieves higher predictive accuracy and better fitting performance, effectively capturing the fluctuation trends of wind power generation. The highlighted section reveals specific areas where the predicted values align well with real data, particularly around samples 30–50, indicating the model’s robustness in capturing fluctuations. In contrast, the traditional BP model shows significant deviations, underscoring its limitations and need for optimization. Overall, the analysis depicts SZCOA-BP as a superior model for wind power forecasting, effectively addressing challenges in prediction accuracy.

In addition, the relative error fluctuation curves for each sample point across different prediction methods are illustrated in Figure 9. This complements the findings of Figure 8 by detailing the relative errors associated with each forecasting model. The upper graph presents a comprehensive view of how each model’s predictions deviate from actual values over the sample range, with emphasis on the stability of these errors. Notably, the SZCOA-BP model maintains low volatility in relative error throughout the trial, demonstrating superior stability compared to its counterparts. The BWO-BP model shows slight increases at certain points but overall performs reliably. In contrast, the original BP model displays significant fluctuations, indicating lower reliability in its predictions. The subplots provide a deeper insight into specific models, with COA-BP and PDO-BP exhibiting higher relative errors, thus reaffirming the necessity for optimizing these models. The analysis shows that while some models can stabilize and produce acceptable predictions, the SZCOA-BP model distinctly outperforms the others by maintaining minimal relative errors, confirming its efficacy in wind power forecasting.

To further verify the accuracy and reliability of the constructed prediction model, we utilized five metrics to assess the predictive performance of the models: mean absolute error (MAE), Mean Absolute Percentage Error (MAPE), mean squared error (MSE), Root Mean Squared Error (RMSE), and the Coefficient of Determination ($R^{2}$) [33,34]. The formulas for each metric are as follows:(26)$$\mathrm{MAE}=\frac{1}{n}\sum_{i=1}^{n}\left|f_{i}-y_{i}\right|$$(27)$$\mathrm{MAPE}=\frac{1}{n}\sum_{i=1}^{n}\left|\frac{f_{i}-y_{i}}{y_{i}}\right|\times 100\%$$(28)$$\mathrm{MSE}=\frac{1}{n}\sum_{i=1}^{n}{\left(f_{i}-y_{i}\right)}^{2}$$(29)$$\mathrm{RMSE}=\sqrt{\mathrm{MSE}}$$(30)$$R^{2}=1-\frac{\sum_{i=1}^{n}{\left(y_{i}-f_{i}\right)}^{2}}{\sum_{i=1}^{n}{\left(y_{i}-\bar{y}\right)}^{2}}$$
where *n* represents the number of samples, $f_{i}$ denotes the predicted value, $y_{i}$ indicates the actual value, and $\bar{y}$ is the mean of the actual values. In general, lower values of the MAE, MMAPE, RMSE, and MSE are preferred, while a value of $R^{2}$ that is closer to 1 is considered better.

When evaluating the predictive performance of the models, we primarily concentrate on MAE and $R^{2}$. MAE measures the difference between predicted and actual values, offering a clear assessment of prediction accuracy. Meanwhile, $R^{2}$ indicates the model’s capacity to account for the variability in the data, serving as a gauge for the quality of the model fit. By integrating these two metrics, we obtain a well-rounded view of model performance, taking into account both the absolute size of prediction errors and the model’s ability to explain data variability.

The MAE and *R*² values for each model, as shown in Figure 10, reveal significant differences that reflect their varying predictive performances. The SZCOA-BP model achieves an *R*² of 94.437% and an MAE of 10.948, indicating extremely high accuracy and reliability in data fitting. This exceptional performance can be attributed to the advantages of the SZCOA-BP model in parameter optimization and structural design, which enable it to capture underlying patterns in the data more effectively. In contrast, the BWO-BP model has an *R*² of 89.562% and an MAE of 12.472. While it performs well, it still falls short of the SZCOA-BP model, suggesting that it has not fully leveraged data features in certain cases. The FLO-BP model, with an *R*² of 88.555% and an MAE of 14.777, demonstrates limitations in its predictive capability, possibly due to its algorithmic complexity and sensitivity to data noise. The *R*² values for the COA-BP and PDO-BP models are 88.232% and 87.558%, with MAE values of 15.256 and 16.045, respectively, indicating that these models perform worse than the aforementioned models when handling complex data. Finally, the original BP model has the lowest *R*² at only 81.167% and an MAE of 18.891, highlighting its inadequacy in predictive tasks, likely due to its simple structure and lack of effective optimization strategies.

The SZCOA-BP model demonstrates the best overall performance in this experiment, with a high *R*² value and low mean absolute error (MAE), indicating its applicability and reliability in practical applications. Although other models perform well in certain aspects, they still require further optimization to enhance their predictive capabilities.

Furthermore, in our evaluation of model performance, we selected MAE, MAPE, and RMSE as the primary assessment criteria. This choice is grounded in the widespread use of these metrics in model evaluation, each offering distinct advantages. MAE provides an intuitive understanding of prediction errors, effectively reflecting the model’s performance in practical applications. MAPE allows for the comparison of predictive accuracy across datasets of varying scales by expressing errors as a percentage, while RMSE highlights the impact of larger errors, making it particularly useful in scenarios where extreme prediction errors are a concern. By employing these three metrics, we can thoroughly assess the predictive capabilities of the models.

To illustrate this assessment, we present a comparison of different models based on MAE, MAPE, and RMSE in Figure 11. To enhance clarity, we display MAE and RMSE values—typically above 10—on the xz-plane, while MAPE values—generally below 5—are shown on the xy-plane, each with appropriate scaling. This arrangement facilitates a more intuitive observation and analysis of the performance differences among the various models.

In this study, the BP model exhibits high values for MAE, MAPE, and RMSE, reflecting its inadequate predictive capability in practical applications. This indicates that the BP model has certain limitations in data fitting and predictive accuracy, especially when dealing with complex nonlinear relationships, where it performs worse than other models. Additionally, the sensitivity of the BP model to parameter settings and training data may affect its performance, particularly in the presence of noise or outliers in the dataset. While the BP model still holds some application value in certain scenarios, its limitations remind us to exercise caution when selecting models. Therefore, optimizing the BP model using metaheuristic algorithms is particularly important.

In contrast, the SZCOA-BP model significantly outperforms the BP model, demonstrating substantial advantages in data fitting and predictive accuracy. This further emphasizes the necessity of improving the BP model. Future research could focus on optimizing the parameter settings of the BP model, introducing more advanced training algorithms, or integrating the strengths of other models to enhance its predictive performance.

Moreover, models such as FLO-BP and BWO-BP perform between the BP and SZCOA-BP models, indicating that these models still possess certain application potential under specific conditions, warranting further exploration and improvement. Through in-depth studies of these models, we can better understand their strengths and weaknesses, thereby providing more effective solutions for practical applications.

The SZCOA-BP model exhibits the lowest error values across all evaluation metrics, highlighting its significant advantages in data fitting and predictive accuracy. This result may be related to the effectiveness of its optimization algorithm and parameter settings, particularly in the MAPE analysis, where the low relative error values further validate its reliability. Therefore, selecting the SZCOA-BP model as the primary predictive tool not only enhances predictive accuracy but also lays a solid foundation for subsequent research.

Finally, in systematically evaluating the performance of different models, we primarily compare five key metrics: MAE, MAPE, MSE, RMSE, and *R*². Figure 12 illustrates the specific performance of these metrics, with the X-axis representing each metric, and each metric accompanied by its corresponding unit for uniform scaling, thereby visually displaying the variations and differences among the metrics. Specifically, the values shown in the figure need to be multiplied by their respective units to obtain the actual values. The Y-axis lists the names of the different models, including SZCOA-BP, BWO-BP, FLO-BP, COA-BP, PDO-BP, and BP. The Z-axis represents the corresponding values of each model under different metrics after uniform scaling.

According to the results shown, the SZCOA-BP model performs excellently across all evaluation metrics, particularly in the Coefficient of Determination (*R*²), where its value significantly exceeds that of other models, indicating that this model can effectively explain the variability of the data and possesses strong fitting capabilities. Specifically, the MAE and RMSE values of the SZCOA-BP model are also relatively low, demonstrating its ability to maintain small absolute errors in practical predictions, making it suitable for applications requiring high precision.

In contrast, the BWO-BP and FLO-BP models also exhibit low MAE and RMSE values, indicating good control over absolute errors during predictions, making them suitable for medium-precision forecasting tasks. However, the performance of the COA-BP and PDO-BP models is relatively inferior, particularly in the MSE and MAPE metrics, which show larger predictive errors that could lead to significant deviations in practical applications. This phenomenon suggests that caution is warranted when selecting models, especially in cases where high predictive accuracy is required.

Additionally, the BP model has the lowest values across all metrics, indicating poor performance in this experiment, likely due to its simple structure or improper parameter settings, which hinder its ability to effectively capture complex patterns in the data. Therefore, it is recommended that future research focus on further optimizing and adjusting the BP model to enhance its performance.

In summary, the SZCOA-BP model outperforms other models in terms of accuracy and stability, making it the recommended best choice for this study. Future research could explore more complex model architectures or ensemble methods to further improve predictive accuracy and the model’s generalization ability. Through in-depth analysis of different models, we can provide more reliable decision support for practical applications.

A detailed comparison of the predictive performance of different models, based on the data in Table 5, reveals significant differences in their accuracy and reliability. First, the SZCOA-BP model excels in MAE, with a value of 10.948, far lower than the 14.777 of FLO-BP and 12.472 of BWO-BP, demonstrating the SZCOA-BP’s exceptional ability to capture actual data trends. This difference indicates that the SZCOA-BP can more effectively reduce prediction errors and enhance the overall accuracy of the model.

In terms of MAPE, the SZCOA-BP’s value of 1.542 is also outstanding, significantly lower than the 3.4731 of COA-BP and 3.5506 of FLO-BP. This result suggests that the SZCOA-BP better reflects actual conditions when processing data, reducing relative errors and enhancing the model’s stability.

For MSE, the SZCOA-BP’s MSE is 233.43, clearly superior to the 790.23 of the standard BP and 480.21 of FLO-BP. This difference emphasizes the SZCOA-BP’s advantage in data fitting capability, effectively lowering prediction errors. In comparison, the BWO-BP’s MSE is 437.98, which, while performing well, still cannot match the SZCOA-BP.

In terms of RMSE, the SZCOA-BP’s value of 15.278 again demonstrates its superiority, being lower than the 22.848 of PDO-BP and 28.111 of the standard BP. This indicates that the SZCOA-BP has a clear advantage in predictive accuracy, accurately reflecting changes in the data.

For time, the BP (backpropagation) model exhibits a rapid convergence time of just 2 min; however, its accuracy on complex datasets is limited to 81.167%, rendering it less suitable for high-precision applications. The COA-BP model and SZCOA-BP model show improved convergence times of 5 min and 6 min, respectively, while achieving higher accuracies of 89.93% and 94.437%. Notably, the SZCOA-BP model stands out as it effectively navigates complex environments, making it an excellent choice for high-precision tasks such as wind power forecasting. While the FLO-BP and BWO-BP models require longer convergence times of 8 min and 10.5 min, respectively, their accuracies do not exceed that of the SZCOA-BP model, indicating that increased efficiency does not necessarily correlate with significant accuracy enhancements. The PDO-BP model, which has the longest convergence time of 11 min, achieves an accuracy of 87.558%, further demonstrating the limitations of this algorithm class.

Finally, regarding *R*², the SZCOA-BP achieves an *R*² of 0.94437, far exceeding other models such as FLO-BP at 0.88555 and BWO-BP at 0.89562. This metric indicates that the SZCOA-BP has a stronger ability to explain data variability, better capturing the underlying patterns in the data.

In conclusion, the SZCOA-BP model performs excellently across all metrics, particularly demonstrating significant advantages in MAE, MAPE, and *R*², indicating its reliability and accuracy in practical applications. In contrast, the standard BP model lags behind on multiple metrics, revealing its inadequacies in predictive capability. Although the PDO-BP, FLO-BP, and BWO-BP models perform well, they still require optimization to reach the level of the SZCOA-BP. Thus, it is evident that the SZCOA-BP model exhibits exceptional performance in predicting wind power generation, significantly outperforming other comparison models, primarily due to the effectiveness of the SZCOA in optimizing the hyperparameters of the BP neural network. Future research could further explore improvements to the SZCOA and apply it to more predictive domains to realize its potential.

#### 3.2.3. Statistical Validation

To enhance the credibility of our findings regarding the performance of the SZCOA-BP model, we conducted a comprehensive statistical analysis. The key components of our analysis are detailed below.
Statistical Validation ImplementationWe systematically validated the performance of the SZCOA-BP model through independent statistical tests. This was achieved using *t*-tests to assess performance differences between the SZCOA-BP model and several benchmark models: standard BP, COA-BP, BWO-BP, FLO-BP, and PDO-BP.*t*-Test AnalysisUtilizing the SciPy library in Python 3.9, we organized and analyzed the *R*² data through *t*-tests. The results of these comparisons are presented in Table 6.Summary of ResultsThe statistical analysis demonstrated that all *p*-values obtained were below the significance threshold of 0.05, indicating a statistically significant performance advantage of the SZCOA-BP model over the other models assessed. For instance, a *p*-value of 0.0003 in the comparison with the standard BP model indicates substantial improvement in wind energy forecasting. The *R*² value of the SZCOA-BP model (94.437) significantly exceeds that of the standard BP model (81.167), reinforcing its practical effectiveness.

#### 3.2.4. Cause Analysis

The advantages of the SZCOA-BP model can be attributed to three improvement strategies we implemented:Improvement of Population Position Update StrategyThis strategy draws inspiration from the Wild Horse Optimization algorithm (WHO) and aims to enhance the algorithm’s global search capability. By dynamically updating the positions of raccoons, the algorithm can explore a broader search space, avoiding premature convergence due to local searches. In traditional optimization algorithms, populations may quickly concentrate on a local optimum, preventing the discovery of better solutions. The SZCOA introduces randomness and a flexible updating mechanism, allowing the population to move over a larger range, thereby increasing the chances of finding the global optimum.Enhanced Ability to Escape Local OptimaInspired by the keen sense of smell of raccoons, this strategy enables the algorithm to promptly perceive changes in the surrounding environment when faced with complex multimodal problems. By simulating the raccoon’s response to predator scents, the algorithm can effectively adjust its search near local optima. In multimodal optimization problems, algorithms often become trapped in local optima. The scent-tracking strategy introduces an environmental perception mechanism, allowing the raccoon to quickly adjust its search direction upon sensing potential threats, thereby effectively escaping local optima and increasing the probability of finding the global optimum.Optimization of Individual Update MechanismThis strategy simulates the growth characteristics of frost particles, utilizing their strong randomness and coverage to enable the algorithm to quickly cover the entire search space. This approach allows the algorithm not only to rapidly identify potential high-quality solutions but also to maintain high precision during the search process. The introduction of the soft frost search strategy enables the algorithm to explore different areas more effectively during the search, avoiding inefficiencies caused by local searches. Additionally, as the number of iterations increases, the algorithm can gradually converge to better solutions, enhancing the overall search accuracy.

These improvements effectively enhance the optimization capability of the SZCOA, allowing it to find better hyperparameter combinations when training the BP neural network, thereby improving the model’s predictive performance.

## 4. Conclusions

In summary, this study presents the SZCOA-BP model, a novel approach that integrates the Coati Optimization Algorithm with backpropagation neural networks to improve wind power forecasting. By employing innovative strategies, such as population position updates and odor tracking, the SZCOA demonstrates enhanced global search capabilities and convergence speed. Experimental results indicate that the SZCOA-BP model outperforms traditional forecasting models, with an *R*² value of 94.4% and a mean absolute error (MAE) of 10.948. This superior performance underscores the model’s effectiveness in accurately predicting wind power generation, contributing to the optimization of energy management systems.

Future research will focus on further optimizing the SZCOA to improve its adaptability in high-dimensional contexts and exploring its application in various predictive domains.

## Figures and Tables

**Figure 1 sensors-25-02438-f001:**
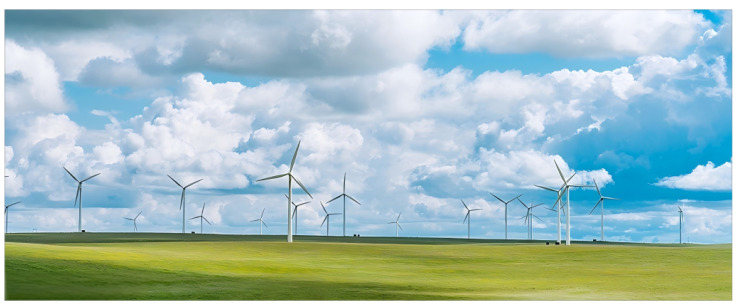
Wind farm cluster on the prairie.

**Figure 2 sensors-25-02438-f002:**
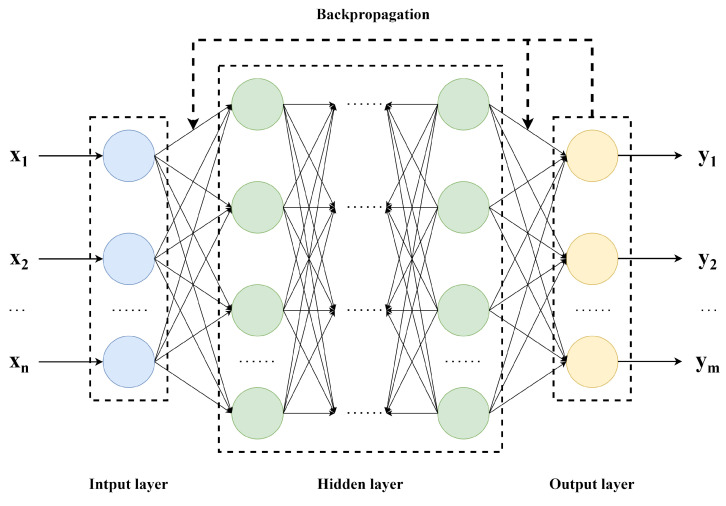
The BP neural network structure diagram.

**Figure 3 sensors-25-02438-f003:**
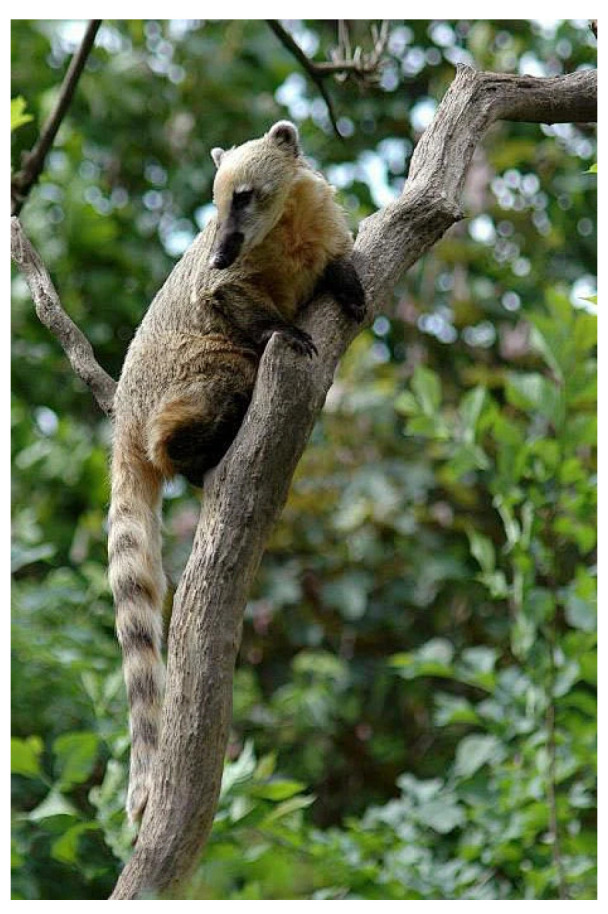
The posture of the coati in the tree.

**Figure 4 sensors-25-02438-f004:**
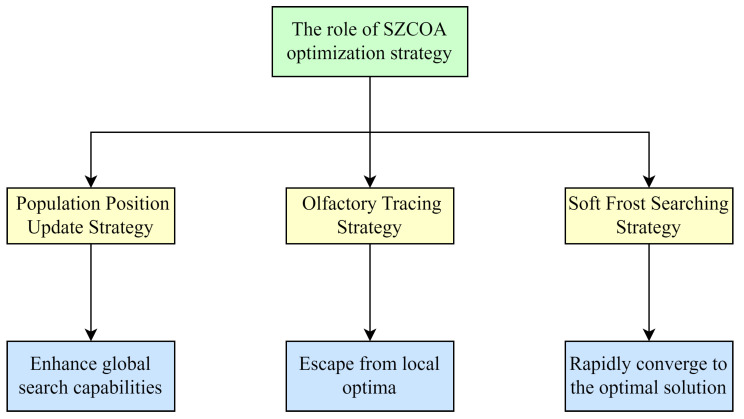
Visual representation of key mechanisms.

**Figure 5 sensors-25-02438-f005:**
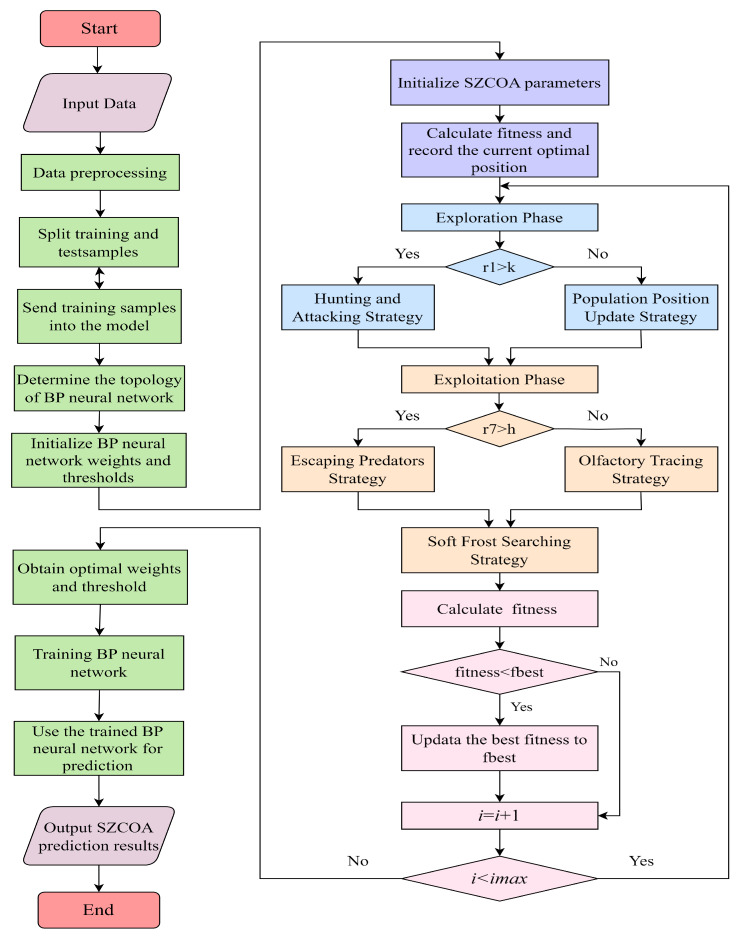
Flowchart of the SZCOA-BP model.

**Figure 6 sensors-25-02438-f006:**
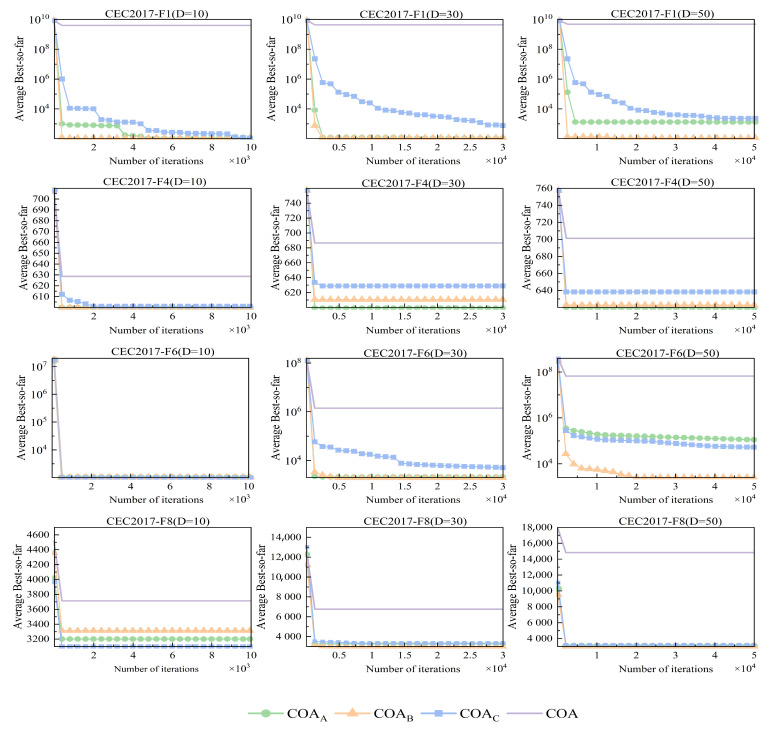
Convergence speed comparison of COA and its improved strategies on F1, F4, F6, and F8.

**Figure 7 sensors-25-02438-f007:**
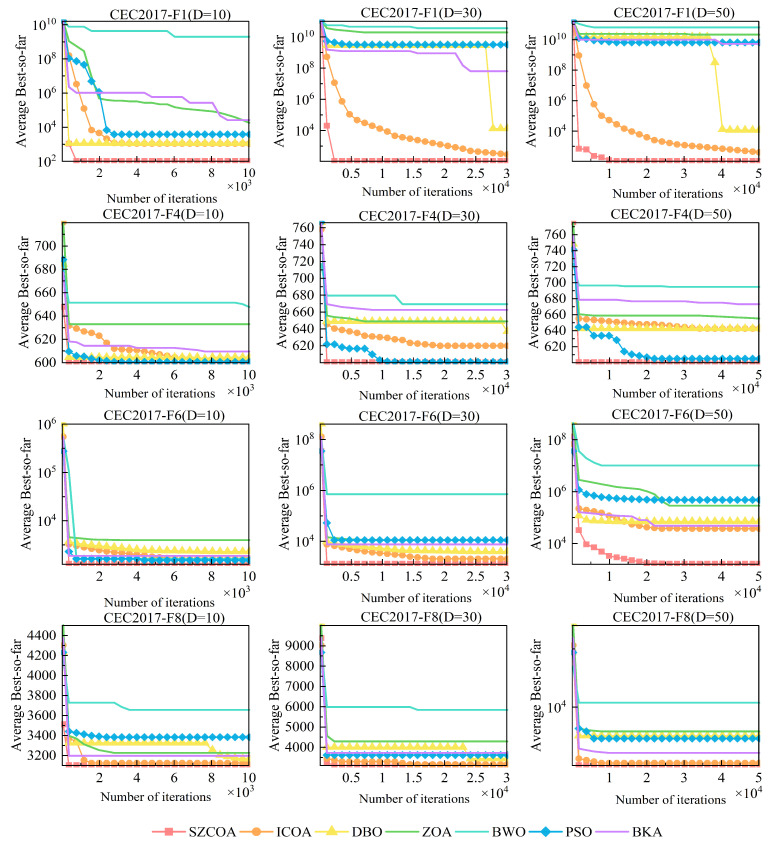
Convergence speed comparison of SZCOA and classical metaheuristics on F1, F4, F6, and F8.

**Figure 8 sensors-25-02438-f008:**
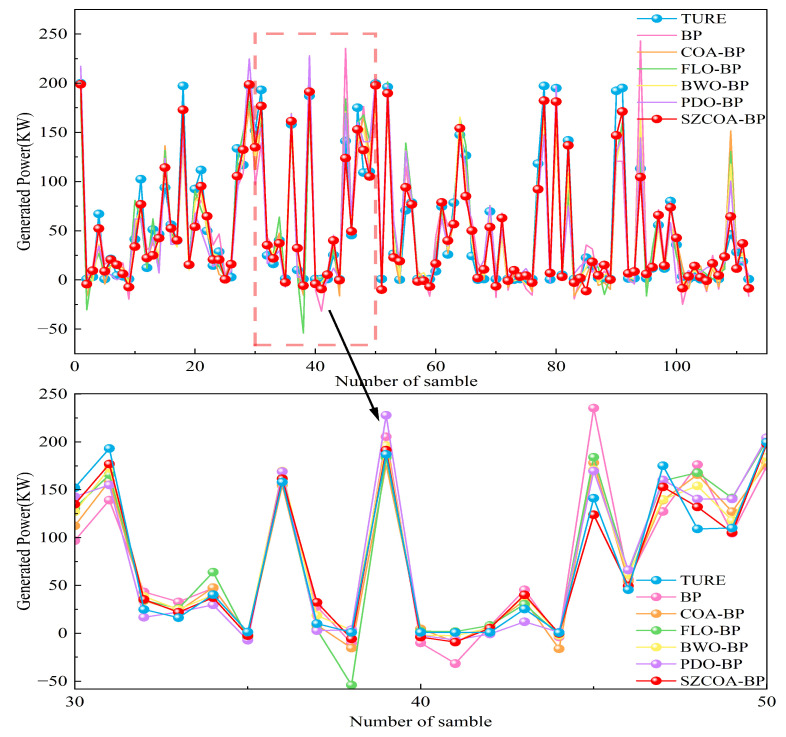
Comparison of predicted power generation values of each model with actual values.

**Figure 9 sensors-25-02438-f009:**
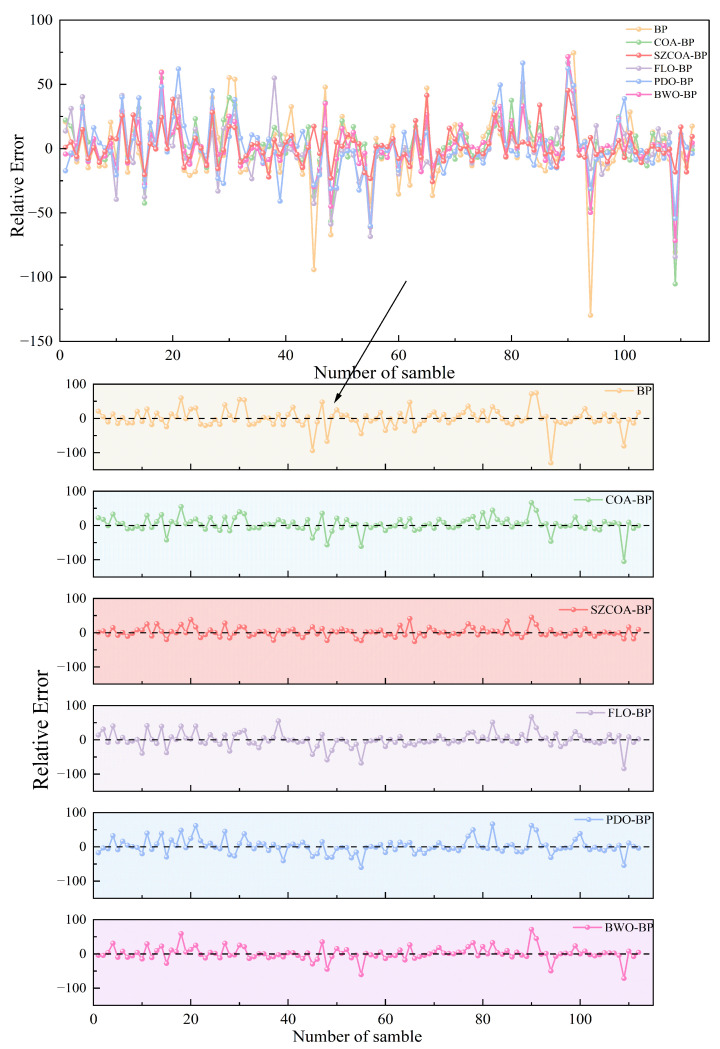
Comparison of relative errors among models.

**Figure 10 sensors-25-02438-f010:**
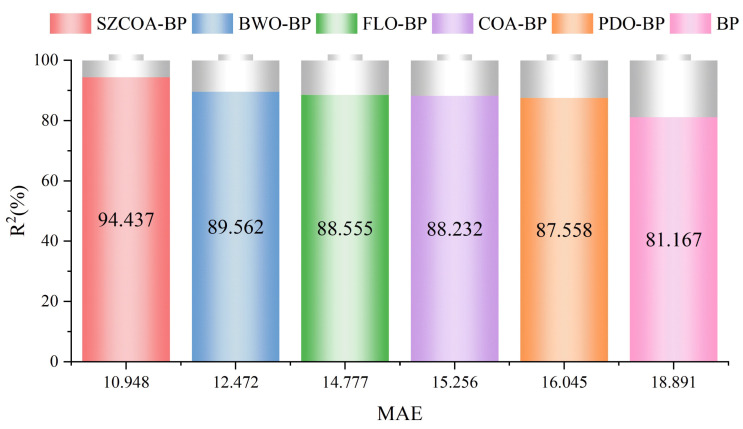
Comparison of *R*² and MAE for different models.

**Figure 11 sensors-25-02438-f011:**
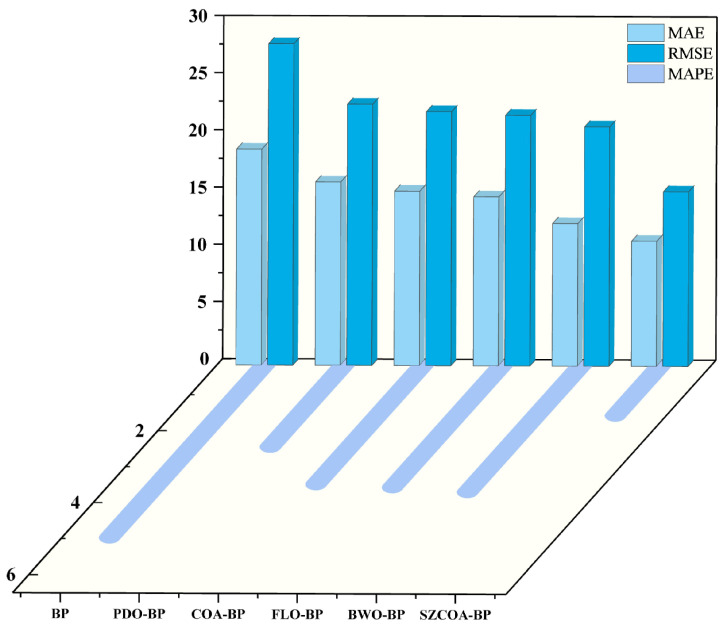
Comparison of MAE, RMSE, and MAPE for different models.

**Figure 12 sensors-25-02438-f012:**
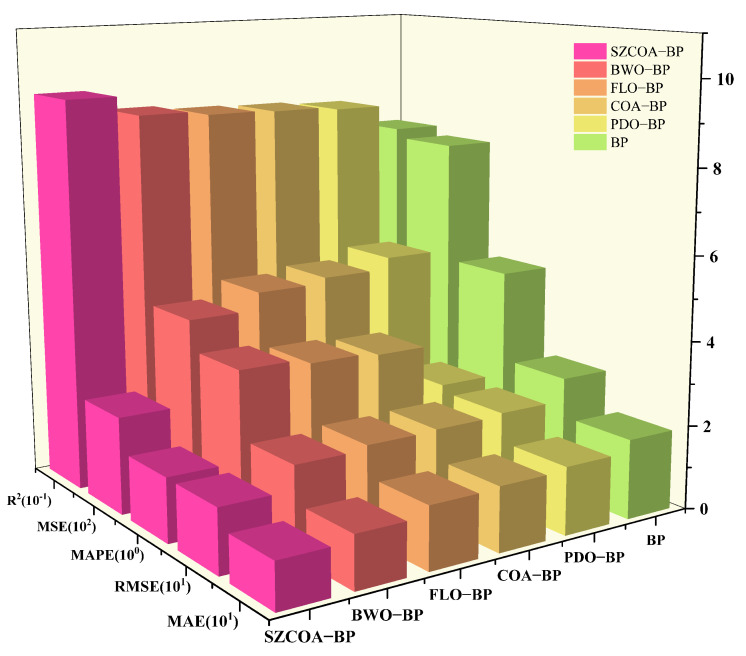
Comparison of evaluation indexes of different models.

**Table 1 sensors-25-02438-t001:** Specific information of the CEC’17 test functions.

No.	Functions	Dim	Best Value
F1	Shifted and Rotated Bent Cigar Function	10/30/50	100
F2	Shifted and Rotated Zakharov Function	10/30/50	300
F3	Shifted and Rotated Rosenbrock’s Function	10/30/50	400
F4	Shifted and Rotated Expanded Scaffer’s F6 Function	10/30/50	600
F5	Hybrid Function 1 (N=3)	10/30/50	1100
F6	Hybrid Function 4 (N=4)	10/30/50	1400
F7	Composition Function 2 (N=3)	10/30/50	2200
F8	Composition Function 6 (N=6)	10/30/50	2800

Search Range: ${\left[-100,\;100\right]}^{D}$.

**Table 2 sensors-25-02438-t002:** Comparison of *COA* and its improved strategies’ solutions on different dimensions of F1–F8.

Function	Dim	Indicator	COA	${\mathrm{COA}}_{A}$	${\mathrm{COA}}_{B}$	${\mathrm{COA}}_{C}$
F1	10	Best	$4.28\times 10^{9}$	** $1.00\times 10^{2}$ **	** $1.00\times 10^{2}$ **	$1.05\times 10^{2}$
		Average	$9.67\times 10^{9}$	$4.03\times 10^{3}$	** $1.00\times 10^{2}$ **	$2.25\times 10^{3}$
		STD	$3.57\times 10^{9}$	$4.10\times 10^{3}$	** $1.53\times 10^{-10}$ **	$2.67\times 10^{3}$
	30	Best	$3.89\times 10^{10}$	** $1.00\times 10^{2}$ **	** $1.00\times 10^{2}$ **	$4.53\times 10^{2}$
		Average	$5.65\times 10^{10}$	$2.51\times 10^{3}$	** $1.00\times 10^{2}$ **	$2.41\times 10^{3}$
		STD	$8.17\times 10^{9}$	$3.90\times 10^{3}$	** $1.01\times 10^{-8}$ **	$2.30\times 10^{3}$
	50	Best	$8.84\times 10^{10}$	$1.81\times 10^{2}$	** $1.00\times 10^{2}$ **	$1.12\times 10^{3}$
		Average	$1.11\times 10^{11}$	$7.02\times 10^{3}$	** $2.61\times 10^{2}$ **	$4.21\times 10^{3}$
		STD	$9.78\times 10^{9}$	$8.43\times 10^{3}$	** $4.35\times 10^{2}$ **	$3.07\times 10^{3}$
F2	10	Best	$5.24\times 10^{3}$	** $3.00\times 10^{2}$ **	** $3.00\times 10^{2}$ **	** $3.00\times 10^{2}$ **
		Average	$1.08\times 10^{4}$	** $3.00\times 10^{2}$ **	** $3.00\times 10^{2}$ **	** $3.00\times 10^{2}$ **
		STD	$2.65\times 10^{3}$	** $1.17\times 10^{-13}$ **	$2.40\times 10^{-12}$	$5.53\times 10^{-5}$
	30	Best	$6.12\times 10^{4}$	** $3.00\times 10^{2}$ **	** $3.00\times 10^{2}$ **	** $3.00\times 10^{2}$ **
		Average	$8.01\times 10^{4}$	$4.11\times 10^{2}$	** $3.00\times 10^{2}$ **	** $3.00\times 10^{2}$ **
		STD	$7.06\times 10^{3}$	$3.82\times 10^{2}$	** $1.39\times 10^{-8}$ **	$2.91\times 10^{-2}$
	50	Best	$1.50\times 10^{5}$	$5.34\times 10^{2}$	** $3.00\times 10^{2}$ **	** $3.00\times 10^{2}$ **
		Average	$1.88\times 10^{5}$	$9.20\times 10^{3}$	** $3.00\times 10^{2}$ **	** $3.00\times 10^{2}$ **
		STD	$1.62\times 10^{4}$	$8.61\times 10^{3}$	** $2.11\times 10^{-3}$ **	$1.41\times 10^{-1}$
F3	10	Best	$6.54\times 10^{2}$	** $4.00\times 10^{2}$ **	** $4.00\times 10^{2}$ **	** $4.00\times 10^{2}$ **
		Average	$1.23\times 10^{3}$	$4.01\times 10^{2}$	** $4.00\times 10^{2}$ **	$4.02\times 10^{2}$
		STD	$6.03\times 10^{2}$	$1.01\times 10^{1}$	** $1.78\times 10^{-11}$ **	$1.10\times 10^{0}$
	30	Best	$9.13\times 10^{3}$	** $4.00\times 10^{2}$ **	** $4.00\times 10^{2}$ **	$4.66\times 10^{2}$
		Average	$1.57\times 10^{4}$	$4.69\times 10^{2}$	** $4.16\times 10^{2}$ **	$4.88\times 10^{2}$
		STD	$2.89\times 10^{3}$	$3.70\times 10^{1}$	$2.86\times 10^{1}$	** $2.25\times 10^{1}$ **
	50	Best	$2.56\times 10^{4}$	$4.23\times 10^{2}$	** $4.00\times 10^{2}$ **	$4.29\times 10^{2}$
		Average	$3.89\times 10^{4}$	$4.95\times 10^{2}$	** $4.37\times 10^{2}$ **	$5.48\times 10^{2}$
		STD	$5.69\times 10^{3}$	$5.21\times 10^{1}$	** $4.60\times 10^{1}$ **	$5.23\times 10^{1}$
F4	10	Best	$6.35\times 10^{2}$	** $6.00\times 10^{2}$ **	** $6.00\times 10^{2}$ **	** $6.00\times 10^{2}$ **
		Average	$6.50\times 10^{2}$	** $6.00\times 10^{2}$ **	** $6.00\times 10^{2}$ **	$6.09\times 10^{2}$
		STD	$8.37\times 10^{0}$	$1.40\times 10^{0}$	** $7.38\times 10^{-1}$ **	$8.35\times 10^{0}$
	30	Best	$6.71\times 10^{2}$	** $6.02\times 10^{2}$ **	$6.03\times 10^{2}$	$6.22\times 10^{2}$
		Average	$6.91\times 10^{2}$	** $6.11\times 10^{2}$ **	$6.15\times 10^{2}$	$6.46\times 10^{2}$
		STD	$7.15\times 10^{0}$	** $6.52\times 10^{0}$ **	$7.01\times 10^{0}$	$1.00\times 10^{1}$
	50	Best	$6.92\times 10^{2}$	** $6.06\times 10^{2}$ **	$6.12\times 10^{2}$	$6.37\times 10^{2}$
		Average	$7.02\times 10^{2}$	** $6.22\times 10^{2}$ **	$6.26\times 10^{2}$	$6.50\times 10^{2}$
		STD	** $5.37\times 10^{0}$ **	$7.81\times 10^{0}$	$6.17\times 10^{0}$	$7.41\times 10^{0}$
F5	10	Best	$1.39\times 10^{3}$	** $1.10\times 10^{3}$ **	** $1.10\times 10^{3}$ **	$1.11\times 10^{3}$
		Average	$2.34\times 10^{3}$	$1.15\times 10^{3}$	** $1.11\times 10^{3}$ **	$1.17\times 10^{3}$
		STD	$1.54\times 10^{3}$	$4.61\times 10^{1}$	** $9.91\times 10^{0}$ **	$6.40\times 10^{1}$
	30	Best	$4.81\times 10^{3}$	$1.17\times 10^{3}$	** $1.13\times 10^{3}$ **	$1.41\times 10^{3}$
		Average	$8.58\times 10^{3}$	$1.28\times 10^{3}$	** $1.20\times 10^{3}$ **	$1.23\times 10^{3}$
		STD	$1.79\times 10^{3}$	$7.35\times 10^{1}$	** $3.66\times 10^{1}$ **	$4.71\times 10^{1}$
	50	Best	$1.94\times 10^{4}$	$1.22\times 10^{3}$	** $1.20\times 10^{3}$ **	$1.23\times 10^{3}$
		Average	$2.57\times 10^{4}$	$1.34\times 10^{3}$	** $1.31\times 10^{3}$ **	$1.34\times 10^{3}$
		STD	$2.81\times 10^{3}$	$6.87\times 10^{1}$	$5.81\times 10^{1}$	** $5.80\times 10^{1}$ **
F6	10	Best	$1.49\times 10^{3}$	$1.42\times 10^{3}$	** $1.40\times 10^{3}$ **	$1.43\times 10^{3}$
		Average	$1.52\times 10^{3}$	$1.46\times 10^{3}$	** $1.43\times 10^{3}$ **	$1.47\times 10^{3}$
		STD	** $2.01\times 10^{1}$ **	$2.11\times 10^{1}$	$3.10\times 10^{1}$	$2.91\times 10^{1}$
	30	Best	$1.92\times 10^{5}$	$1.74\times 10^{3}$	** $1.49\times 10^{3}$ **	$1.79\times 10^{3}$
		Average	$3.98\times 10^{6}$	$2.29\times 10^{4}$	** $1.68\times 10^{3}$ **	$6.34\times 10^{3}$
		STD	$2.99\times 10^{6}$	$2.04\times 10^{5}$	** $1.42\times 10^{2}$ **	$5.27\times 10^{3}$
	50	Best	$1.04\times 10^{7}$	$2.42\times 10^{3}$	** $1.77\times 10^{3}$ **	$6.01\times 10^{3}$
		Average	$1.34\times 10^{8}$	$6.34\times 10^{4}$	** $2.20\times 10^{3}$ **	$5.10\times 10^{4}$
		STD	$1.08\times 10^{8}$	$1.74\times 10^{5}$	** $2.63\times 10^{2}$ **	$3.87\times 10^{5}$
F7	10	Best	$2.50\times 10^{3}$	** $2.20\times 10^{3}$ **	$2.23\times 10^{3}$	$2.22\times 10^{3}$
		Average	$3.16\times 10^{3}$	** $2.28\times 10^{3}$ **	$2.30\times 10^{3}$	$2.32\times 10^{3}$
		STD	$3.82\times 10^{2}$	$2.93\times 10^{1}$	** $1.24\times 10^{1}$ **	$1.52\times 10^{2}$
	30	Best	$7.39\times 10^{3}$	** $2.30\times 10^{3}$ **	** $2.30\times 10^{3}$ **	** $2.30\times 10^{3}$ **
		Average	$9.46\times 10^{3}$	$2.41\times 10^{3}$	** $2.30\times 10^{3}$ **	$3.51\times 10^{3}$
		STD	$7.30\times 10^{2}$	$7.23\times 10^{2}$	** $3.11\times 10^{0}$ **	$2.04\times 10^{3}$
	50	Best	$1.60\times 10^{4}$	** $2.30\times 10^{3}$ **	** $2.30\times 10^{3}$ **	$9.11\times 10^{3}$
		Average	$1.69\times 10^{4}$	** $8.53\times 10^{3}$ **	$9.12\times 10^{3}$	$1.05\times 10^{4}$
		STD	$1.10\times 10^{3}$	$3.37\times 10^{3}$	$2.05\times 10^{3}$	** $8.00\times 10^{2}$ **
F8	10	Best	$3.43\times 10^{3}$	** $3.10\times 10^{3}$ **	** $3.10\times 10^{3}$ **	** $3.10\times 10^{3}$ **
		Average	$3.73\times 10^{3}$	$3.41\times 10^{3}$	$3.36\times 10^{3}$	** $3.30\times 10^{3}$ **
		STD	$1.59\times 10^{2}$	$1.54\times 10^{2}$	$1.30\times 10^{2}$	** $1.27\times 10^{2}$ **
	30	Best	$5.62\times 10^{3}$	** $3.10\times 10^{3}$ **	** $3.10\times 10^{3}$ **	** $3.10\times 10^{3}$ **
		Average	$7.53\times 10^{3}$	$3.21\times 10^{3}$	** $3.13\times 10^{3}$ **	$3.18\times 10^{3}$
		STD	$7.12\times 10^{2}$	$9.00\times 10^{1}$	$5.63\times 10^{1}$	** $5.51\times 10^{1}$ **
	50	Best	$1.03\times 10^{4}$	** $3.25\times 10^{3}$ **	$3.26\times 10^{3}$	** $3.25\times 10^{3}$ **
		Average	$1.39\times 10^{4}$	$3.30\times 10^{3}$	** $3.29\times 10^{3}$ **	$3.30\times 10^{3}$
		STD	$1.51\times 10^{3}$	$1.61\times 10^{1}$	** $1.41\times 10^{1}$ **	$2.00\times 10^{1}$

**Note**: The bold values represent the optimal values found under different metrics.

**Table 3 sensors-25-02438-t003:** Comparison of optimization algorithms’ solutions on different functions and dimensions.

Function	Dim	Indicator	SZCOA	ICOA	DBO	ZOA	BWO	PSO	BKA
F1	10	Best	** $1.00\times 10^{2}$ **	$1.10\times 10^{2}$	$5.56\times 10^{2}$	$1.09\times 10^{2}$	$2.38\times 10^{9}$	$1.43\times 10^{2}$	$1.30\times 10^{4}$
		Average	** $1.00\times 10^{2}$ **	$1.09\times 10^{3}$	$6.97\times 10^{3}$	$4.76\times 10^{8}$	$4.65\times 10^{9}$	$3.78\times 10^{7}$	$1.84\times 10^{8}$
		STD	** $5.46\times 10^{-11}$ **	$1.36\times 10^{3}$	$4.28\times 10^{3}$	$6.22\times 10^{8}$	$1.42\times 10^{9}$	$2.07\times 10^{8}$	$5.46\times 10^{8}$
	30	Best	** $1.00\times 10^{2}$ **	$1.22\times 10^{2}$	$1.42\times 10^{2}$	$1.21\times 10^{9}$	$3.59\times 10^{10}$	$2.48\times 10^{2}$	$2.62\times 10^{6}$
		Average	** $1.00\times 10^{2}$ **	$8.28\times 10^{2}$	$7.78\times 10^{3}$	$9.23\times 10^{9}$	$4.15\times 10^{10}$	$3.26\times 10^{9}$	$1.42\times 10^{9}$
		STD	** $2.82\times 10^{-7}$ **	$6.54\times 10^{2}$	$7.60\times 10^{3}$	$4.21\times 10^{9}$	$3.04\times 10^{9}$	$3.86\times 10^{9}$	$4.78\times 10^{9}$
	50	Best	** $1.00\times 10^{2}$ **	$3.12\times 10^{2}$	$2.43\times 10^{2}$	$4.31\times 10^{9}$	$8.43\times 10^{10}$	$1.53\times 10^{4}$	$5.93\times 10^{8}$
		Average	** $1.00\times 10^{2}$ **	$6.12\times 10^{2}$	$4.51\times 10^{4}$	$2.53\times 10^{10}$	$9.22\times 10^{10}$	$7.22\times 10^{9}$	$5.87\times 10^{9}$
		STD	** $4.18\times 10^{-3}$ **	$1.50\times 10^{2}$	$1.06\times 10^{5}$	$8.73\times 10^{9}$	$4.19\times 10^{9}$	$6.14\times 10^{9}$	$4.38\times 10^{9}$
F2	10	Best	** $3.00\times 10^{2}$ **	** $3.00\times 10^{2}$ **	** $3.00\times 10^{2}$ **	** $3.00\times 10^{2}$ **	$4.77\times 10^{3}$	** $3.00\times 10^{2}$ **	** $3.00\times 10^{2}$ **
		Average	** $3.00\times 10^{2}$ **	** $3.00\times 10^{2}$ **	** $3.00\times 10^{2}$ **	$1.12\times 10^{3}$	$7.14\times 10^{3}$	** $3.00\times 10^{2}$ **	$1.39\times 10^{3}$
		STD	** $3.58\times 10^{-9}$ **	$4.09\times 10^{-8}$	$9.79\times 10^{-6}$	$1.30\times 10^{3}$	$9.67\times 10^{2}$	$1.06\times 10^{-8}$	$2.85\times 10^{3}$
	30	Best	** $3.00\times 10^{2}$ **	** $3.00\times 10^{2}$ **	** $3.00\times 10^{2}$ **	$4.57\times 10^{3}$	$5.08\times 10^{4}$	** $3.00\times 10^{2}$ **	$3.91\times 10^{2}$
		Average	** $3.00\times 10^{2}$ **	$3.01\times 10^{2}$	$4.17\times 10^{2}$	$1.52\times 10^{4}$	$6.68\times 10^{4}$	$9.07\times 10^{3}$	$1.31\times 10^{4}$
		STD	** $2.27\times 10^{-7}$ **	$6.58\times 10^{-1}$	$1.85\times 10^{2}$	$5.45\times 10^{3}$	$5.39\times 10^{3}$	$1.17\times 10^{4}$	$2.55\times 10^{4}$
	50	Best	** $3.00\times 10^{2}$ **	$1.05\times 10^{3}$	$2.63\times 10^{3}$	$2.09\times 10^{4}$	$1.07\times 10^{5}$	** $3.00\times 10^{2}$ **	$8.12\times 10^{3}$
		Average	** $3.00\times 10^{2}$ **	$1.32\times 10^{3}$	$3.60\times 10^{4}$	$4.39\times 10^{4}$	$1.48\times 10^{5}$	$2.56\times 10^{4}$	$4.09\times 10^{4}$
		STD	** $1.03\times 10^{-5}$ **	$3.43\times 10^{2}$	$3.19\times 10^{4}$	$1.16\times 10^{4}$	$1.34\times 10^{4}$	$2.71\times 10^{4}$	$4.39\times 10^{4}$
F3	10	Best	** $4.00\times 10^{2}$ **	** $4.00\times 10^{2}$ **	** $4.00\times 10^{2}$ **	$4.02\times 10^{2}$	$5.02\times 10^{2}$	$4.03\times 10^{2}$	** $4.00\times 10^{2}$ **
		Average	** $4.00\times 10^{2}$ **	** $4.00\times 10^{2}$ **	$4.13\times 10^{2}$	$4.41\times 10^{2}$	$6.08\times 10^{2}$	$4.11\times 10^{2}$	$4.01\times 10^{2}$
		STD	** $1.80\times 10^{-8}$ **	$3.60\times 10^{-1}$	$2.18\times 10^{1}$	$4.95\times 10^{1}$	$7.80\times 10^{1}$	$1.32\times 10^{1}$	$1.73\times 10^{0}$
	30	Best	** $4.00\times 10^{2}$ **	** $4.00\times 10^{2}$ **	$4.60\times 10^{2}$	$5.35\times 10^{2}$	$6.11\times 10^{3}$	$4.86\times 10^{2}$	$4.07\times 10^{2}$
		Average	** $4.11\times 10^{2}$ **	$4.43\times 10^{2}$	$5.01\times 10^{2}$	$1.19\times 10^{3}$	$8.69\times 10^{3}$	$6.90\times 10^{2}$	$1.58\times 10^{3}$
		STD	** $1.83\times 10^{1}$ **	$3.53\times 10^{1}$	$2.11\times 10^{1}$	$7.79\times 10^{2}$	$9.99\times 10^{2}$	$3.46\times 10^{2}$	$3.29\times 10^{3}$
	50	Best	** $4.00\times 10^{2}$ **	** $4.00\times 10^{2}$ **	$4.53\times 10^{2}$	$1.15\times 10^{3}$	$2.05\times 10^{4}$	$6.32\times 10^{2}$	$5.22\times 10^{2}$
		Average	** $4.35\times 10^{2}$ **	$4.48\times 10^{2}$	$5.59\times 10^{2}$	$3.56\times 10^{3}$	$2.59\times 10^{4}$	$1.51\times 10^{3}$	$4.97\times 10^{3}$
		STD	** $3.81\times 10^{1}$ **	$8.42\times 10^{1}$	$5.69\times 10^{1}$	$2.02\times 10^{3}$	$2.54\times 10^{3}$	$1.36\times 10^{3}$	$1.11\times 10^{4}$
F4	10	Best	** $6.00\times 10^{2}$ **	** $6.00\times 10^{2}$ **	$6.02\times 10^{2}$	$6.18\times 10^{2}$	$6.30\times 10^{2}$	** $6.00\times 10^{2}$ **	$6.05\times 10^{2}$
		Average	** $6.00\times 10^{2}$ **	$6.01\times 10^{2}$	$6.06\times 10^{2}$	$6.24\times 10^{2}$	$6.38\times 10^{2}$	** $6.00\times 10^{2}$ **	$6.22\times 10^{2}$
		STD	$1.35\times 10^{0}$	$5.23\times 10^{0}$	$5.14\times 10^{0}$	$9.43\times 10^{0}$	$4.31\times 10^{0}$	** $2.47\times 10^{-1}$ **	$7.84\times 10^{0}$
	30	Best	** $6.00\times 10^{2}$ **	$6.18\times 10^{2}$	$6.15\times 10^{2}$	$6.42\times 10^{2}$	$6.69\times 10^{2}$	** $6.00\times 10^{2}$ **	$6.46\times 10^{2}$
		Average	$6.12\times 10^{2}$	$6.30\times 10^{2}$	$6.40\times 10^{2}$	$6.52\times 10^{2}$	$6.80\times 10^{2}$	** $6.02\times 10^{2}$ **	$6.59\times 10^{2}$
		STD	$1.13\times 10^{1}$	$3.12\times 10^{1}$	$1.24\times 10^{1}$	$1.13\times 10^{1}$	$1.13\times 10^{1}$	** $2.43\times 10^{0}$ **	$1.14\times 10^{1}$
	50	Best	** $6.00\times 10^{2}$ **	$6.23\times 10^{2}$	$6.34\times 10^{2}$	$6.51\times 10^{2}$	$6.87\times 10^{2}$	$6.01\times 10^{2}$	$6.61\times 10^{2}$
		Average	$6.31\times 10^{2}$	$6.55\times 10^{2}$	$6.55\times 10^{2}$	$6.59\times 10^{2}$	$6.94\times 10^{2}$	** $6.06\times 10^{2}$ **	$6.70\times 10^{2}$
		STD	$1.53\times 10^{1}$	$2.12\times 10^{1}$	$1.68\times 10^{1}$	$3.61\times 10^{0}$	$4.15\times 10^{0}$	** $3.48\times 10^{0}$ **	$9.23\times 10^{0}$
F5	10	Best	** $1.10\times 10^{3}$ **	$1.11\times 10^{3}$	$1.11\times 10^{3}$	$1.11\times 10^{3}$	$1.28\times 10^{3}$	** $1.10\times 10^{3}$ **	$1.22\times 10^{3}$
		Average	** $1.11\times 10^{3}$ **	$1.12\times 10^{3}$	$1.16\times 10^{3}$	$1.15\times 10^{3}$	$1.56\times 10^{3}$	** $1.11\times 10^{3}$ **	$1.56\times 10^{3}$
		STD	** $3.96\times 10^{0}$ **	$2.48\times 10^{1}$	$6.65\times 10^{1}$	$3.83\times 10^{1}$	$2.07\times 10^{2}$	$1.88\times 10^{1}$	$2.26\times 10^{2}$
	30	Best	** $1.11\times 10^{3}$ **	$1.14\times 10^{3}$	$1.19\times 10^{3}$	$1.31\times 10^{1}$	$3.35\times 10^{3}$	$1.17\times 10^{3}$	$1.18\times 10^{3}$
		Average	** $1.18\times 10^{3}$ **	$1.20\times 10^{3}$	$1.37\times 10^{3}$	$1.92\times 10^{3}$	$5.02\times 10^{3}$	$1.30\times 10^{3}$	$1.73\times 10^{3}$
		STD	** $5.91\times 10^{1}$ **	$6.14\times 10^{1}$	$1.07\times 10^{2}$	$6.90\times 10^{2}$	$7.18\times 10^{2}$	$1.12\times 10^{2}$	$1.31\times 10^{3}$
	50	Best	** $1.25\times 10^{3}$ **	$1.29\times 10^{3}$	$1.32\times 10^{3}$	$1.43\times 10^{3}$	$1.32\times 10^{4}$	$1.41\times 10^{3}$	$1.40\times 10^{3}$
		Average	** $1.33\times 10^{3}$ **	** $1.33\times 10^{3}$ **	$1.58\times 10^{3}$	$4.05\times 10^{3}$	$1.65\times 10^{4}$	$2.42\times 10^{3}$	$5.17\times 10^{3}$
		STD	$6.61\times 10^{1}$	** $3.29\times 10^{1}$ **	$1.35\times 10^{2}$	$2.19\times 10^{3}$	$1.61\times 10^{3}$	$4.59\times 10^{3}$	$6.21\times 10^{3}$
F6	10	Best	** $1.40\times 10^{3}$ **	$1.43\times 10^{3}$	$1.43\times 10^{3}$	$1.45\times 10^{3}$	$1.50\times 10^{3}$	$1.41\times 10^{3}$	$1.43\times 10^{3}$
		Average	** $1.41\times 10^{3}$ **	$1.48\times 10^{3}$	$1.50\times 10^{3}$	$3.40\times 10^{3}$	$1.55\times 10^{3}$	$1.50\times 10^{3}$	$1.47\times 10^{3}$
		STD	** $9.51\times 10^{0}$ **	$3.15\times 10^{1}$	$4.55\times 10^{1}$	$2.05\times 10^{3}$	$2.78\times 10^{1}$	$3.73\times 10^{2}$	$3.21\times 10^{1}$
	30	Best	** $1.42\times 10^{3}$ **	$2.02\times 10^{3}$	$2.09\times 10^{3}$	$1.96\times 10^{3}$	$1.85\times 10^{5}$	$1.83\times 10^{3}$	$1.58\times 10^{3}$
		Average	** $1.48\times 10^{3}$ **	$3.23\times 10^{3}$	$3.29\times 10^{4}$	$1.37\times 10^{5}$	$9.13\times 10^{5}$	$1.91\times 10^{4}$	$8.88\times 10^{3}$
		STD	** $5.12\times 10^{1}$ **	$2.05\times 10^{3}$	$3.88\times 10^{4}$	$3.55\times 10^{5}$	$5.96\times 10^{5}$	$4.26\times 10^{4}$	$2.45\times 10^{4}$
	50	Best	** $1.50\times 10^{3}$ **	$9.51\times 10^{3}$	$5.30\times 10^{3}$	$6.49\times 10^{3}$	$3.95\times 10^{6}$	$1.95\times 10^{4}$	$2.99\times 10^{3}$
		Average	** $1.81\times 10^{3}$ **	$1.23\times 10^{4}$	$3.21\times 10^{5}$	$8.07\times 10^{5}$	$1.35\times 10^{7}$	$4.72\times 10^{5}$	$3.76\times 10^{5}$
		STD	** $1.34\times 10^{2}$ **	$4.62\times 10^{3}$	$2.66\times 10^{5}$	$1.32\times 10^{6}$	$5.28\times 10^{6}$	$7.53\times 10^{5}$	$1.28\times 10^{6}$
F7	10	Best	** $2.20\times 10^{3}$ **	$2.22\times 10^{3}$	$2.25\times 10^{3}$	$2.34\times 10^{3}$	$2.32\times 10^{3}$	$2.30\times 10^{3}$	$2.23\times 10^{3}$
		Average	** $2.28\times 10^{3}$ **	$2.29\times 10^{3}$	$2.31\times 10^{3}$	$2.45\times 10^{3}$	$2.47\times 10^{3}$	$2.33\times 10^{3}$	$2.45\times 10^{3}$
		STD	** $1.93\times 10^{1}$ **	$3.50\times 10^{1}$	$2.08\times 10^{1}$	$6.82\times 10^{1}$	$1.18\times 10^{2}$	$7.91\times 10^{1}$	$1.10\times 10^{2}$
	30	Best	** $2.30\times 10^{3}$ **	** $2.30\times 10^{3}$ **	** $2.30\times 10^{3}$ **	$3.31\times 10^{3}$	$6.21\times 10^{3}$	** $2.30\times 10^{3}$ **	$2.41\times 10^{3}$
		Average	$3.19\times 10^{3}$	** $2.30\times 10^{3}$ **	$3.67\times 10^{3}$	$6.10\times 10^{3}$	$7.45\times 10^{3}$	$4.25\times 10^{3}$	$5.93\times 10^{3}$
		STD	$1.23\times 10^{3}$	** $8.22\times 10^{-3}$ **	$1.99\times 10^{3}$	$8.79\times 10^{2}$	$1.36\times 10^{3}$	$1.40\times 10^{3}$	$1.54\times 10^{3}$
	50	Best	** $6.50\times 10^{3}$ **	$8.29\times 10^{3}$	$1.01\times 10^{4}$	$8.69\times 10^{3}$	$1.47\times 10^{4}$	$6.91\times 10^{3}$	$8.98\times 10^{3}$
		Average	** $7.76\times 10^{3}$ **	$8.85\times 10^{3}$	$1.23\times 10^{4}$	$1.00\times 10^{4}$	$1.57\times 10^{4}$	$8.46\times 10^{3}$	$1.09\times 10^{4}$
		STD	$1.50\times 10^{3}$	$5.25\times 10^{2}$	$6.37\times 10^{2}$	$6.20\times 10^{2}$	** $3.76\times 10^{2}$ **	$9.10\times 10^{2}$	$2.13\times 10^{3}$
F8	10	Best	** $3.10\times 10^{3}$ **	** $3.10\times 10^{3}$ **	** $3.10\times 10^{3}$ **	** $3.10\times 10^{3}$ **	$3.27\times 10^{3}$	** $3.10\times 10^{3}$ **	** $3.10\times 10^{3}$ **
		Average	** $3.25\times 10^{3}$ **	$3.31\times 10^{3}$	$3.34\times 10^{3}$	$3.43\times 10^{3}$	$3.47\times 10^{3}$	$3.34\times 10^{3}$	$3.28\times 10^{3}$
		STD	** $1.25\times 10^{2}$ **	$1.96\times 10^{2}$	$1.36\times 10^{2}$	$1.72\times 10^{2}$	$1.58\times 10^{2}$	$1.38\times 10^{2}$	$1.29\times 10^{2}$
	30	Best	** $3.10\times 10^{3}$ **	** $3.10\times 10^{3}$ **	$3.21\times 10^{3}$	$3.31\times 10^{3}$	$5.39\times 10^{3}$	$3.27\times 10^{3}$	$3.21\times 10^{3}$
		Average	** $3.10\times 10^{3}$ **	** $3.10\times 10^{3}$ **	$3.27\times 10^{3}$	$3.83\times 10^{3}$	$5.73\times 10^{3}$	$3.42\times 10^{3}$	$3.65\times 10^{3}$
		STD	** $5.71\times 10^{-9}$ **	$2.32\times 10^{-5}$	$6.51\times 10^{1}$	$3.43\times 10^{2}$	$1.64\times 10^{2}$	$1.32\times 10^{2}$	$8.42\times 10^{2}$
	50	Best	** $3.23\times 10^{3}$ **	$3.26\times 10^{3}$	$3.31\times 10^{3}$	$4.39\times 10^{3}$	$1.01\times 10^{4}$	$3.45\times 10^{3}$	$3.40\times 10^{3}$
		Average	** $3.28\times 10^{3}$ **	$3.38\times 10^{3}$	$4.57\times 10^{3}$	$5.46\times 10^{3}$	$1.07\times 10^{4}$	$4.56\times 10^{3}$	$4.46\times 10^{3}$
		STD	** $3.32\times 10^{1}$ **	$8.46\times 10^{1}$	$1.74\times 10^{3}$	$6.34\times 10^{2}$	$3.49\times 10^{2}$	$1.08\times 10^{3}$	$2.31\times 10^{3}$

**Note**: The bold values represent the optimal values found under different metrics.

**Table 4 sensors-25-02438-t004:** Sample data related to wind power generation.

No.	10-m Wind Speed of the Wind Measurement Tower/m·s^−1^	⋯	10-m Wind Direction of the Wind Measurement Tower/°	⋯	Temperature /°C	Air Pressure /hPa
1	2.803		214.542		13.155	874.684
2	3.132		209.531		13.117	874.684
3	1.359		219.760		13.085	874.005
⋮	⋮	⋯	⋮	⋯	⋮	⋮
497	4.940		252.930		14.476	871.126
498	4.662		237.509		14.521	870.690
499	3.515		254.262		14.591	870.219
⋮	⋮	⋯	⋮	⋯	⋮	⋮
911	8.877		59.365		13.230	869.825
912	7.479		76.038		13.184	869.994

**Table 5 sensors-25-02438-t005:** Model performance metrics.

Model	MAE	MAPE	MSE	RMSE	$R^{2}$	*T* (min)
BP	18.891	4.9717	790.23	28.111	0.81167	**2**
COA-BP	15.256	3.4731	493.78	22.221	0.88232	5
FLO-BP	14.777	3.5506	480.21	21.914	0.88555	8
BWO-BP	12.472	3.6762	437.98	20.928	0.89562	10.5
PDO-BP	16.045	2.4381	522.05	22.848	0.87558	11
SZCOA-BP	**10.948**	**1.542**	**233.43**	**15.278**	**0.94437**	6

**Note:** The bold values indicate the best performance value among the models.

**Table 6 sensors-25-02438-t006:** *t*-test results comparing the *R*² values of the SZCOA-BP model with benchmark models.

Model	*t*-Statistic	*p*-Value
Standard BP	6.484	0.0003
COA-BP	3.064	0.012
BWO-BP	3.418	0.003
FLO-BP	2.981	0.005
PDO-BP	2.355	0.008

## Data Availability

The statistical data come from the Alibaba Cloud Tianchi dataset.

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
