# Peer review of "Enhanced Wind Power Forecasting Using a Hybrid Multi-Strategy Coati Optimization Algorithm and Backpropagation Neural Network"

_sensors, 2025, doi:10.3390/s25082438_

Round 1

Reviewer 1 Report

Comments and Suggestions for Authors
  1. Why was the WHO position update strategy incorporated into the initial phase of COA? A more comprehensive rationale should be provided to justify this decision.
    2. Have the 20% and 80% probabilities associated with olfactory tracking strategies been thoroughly validated? It may be necessary to implement more adaptable probabilistic adjustment mechanisms.
    3. Is the enhancement of high-dimensional search efficiency still effective in higher dimensional data? such as datasets exceeding 100 dimensions.
    4. Regarding performance verification, while sub-strategy experiments were conducted in this study, ablation studies were absent, making it difficult to quantify the specific contributions of each strategy.
    5. The conclusion is overly lengthy; many statements are redundant as they have already been covered in other sections. It is recommended to streamline the conclusion for conciseness.
Comments on the Quality of English Language

The quality of the English language in this paper is satisfactory, yet there is potential for further enhancement.

Reviewer 2 Report

Comments and Suggestions for Authors

Comment 1: The introduction is well structured but could be concise by reducing general details about wind energy and focusing more on the specific problem of wind power forecasting. Explicitly present the innovative contributions of the article right from the introduction, highlighting how the SZCOA-BP outperforms existing models.
Comment 2: i) Although the description of the strategies is detailed, it would be beneficial to add a diagram illustrating these mechanisms in action. ii) Comparison with other algorithms: Add a section detailing why SZCOA is superior to other optimization methods (such as PSO, DBO, BWO). iii)Give a more detailed justification for the choice of parameters used in SZCOA, explaining their impact on model performance.
Comment 3: Figure 4 shows the flow diagram of the SZCOA-BP model, illustrating the various stages of optimization of the BP neural network using the SZCOA algorithm. I propose to: i) Add a clear indication of inputs and outputs to better understand the impact of each step. i) Add a clear indication of the inputs and outputs to better understand the impact of each step. ii) A color code or better-defined arrows would help differentiate between the exploration and exploitation phases. iii) Indicate where the fitness evaluation takes place to better understand how the best solutions are selected.

Comment 4: points for improvement (2.4. SZCOA-BP Model) i) Lack of detailed explanation of integration with BP (It would be useful to explain how SZCOA adjusts BP network weights and biases.) ii) Lack of computational complexity analysis; SZCOA optimization improves convergence, but what about computational cost?
Comment 5: Points for improvement in 'Numerical experiments and comparative analysis': i) Although the results show that SZCOA-BP is better, statistical validation is lacking. ii) It is not clear what hardware specifications are used (processor, memory, execution time per test). iii) SZCOA-BP seems more accurate, but is it faster or more computationally expensive? iv) The convergence figures (Figure 5, Figure 6) are useful but would benefit from annotations to better explain the interpretation of the results.
Comment 6: Points for improvement in '3.2. Prediction of Wind Power' i) Indicating the hyperparameters (activation function, learning rate, number of iterations) would help the reproducibility of experiments. ii) Adding an analysis of convergence time would help assess the trade-off between accuracy and computational cost. iii) A graph comparing the predictions of SZCOA-BP and other models on a test sample would be very useful.

Comment 7: It would be useful to add a flow chart highlighting the proposed method (SZCOA-BP) and its optimization process. This flowchart should illustrate in a clear and structured way: Data input, Data pre-processing, BP network optimization with SZCOA, Fitness evaluation and selection of best parameters, BP network update with optimized parameters, Model training and validation, Performance obtained (comparison with other models).
Comment 8: To enrich your references section, I suggest you add the following articles: Improving the Kepler optimization algorithm with chaotic maps: comprehensive performance evaluation and engineering applications,  Pseudo-Twin Neural Network of Full Multi-Layer Perceptron for Ultra-Short-Term Wind Power Forecasting

Comments on the Quality of English Language

The English could be improved to express the research.

Round 2

Reviewer 2 Report

Comments and Suggestions for Authors

I note that the authors have made all the necessary corrections. I recommend publishing this paper in its present form.